# Learning Invariant Graph Representations for Out-of-Distribution Generalization

**Haoyang Li,  Ziwei Zhang,  Xin Wang**[*]**,  Wenwu Zhu**[*]
Tsinghua University
lihy18@mails.tsinghua.edu.cn, {zwzhang,xin_wang,wwzhu}@tsinghua.edu.cn

## Abstract

Graph representation learning has shown effectiveness when testing and training graph data come from the same distribution, but most existing approaches fail to generalize under distribution shifts. Invariant learning, backed by the invariance principle from causality, can achieve guaranteed generalization under distribution shifts in theory and has shown great successes in practice. However, invariant learning for graphs under distribution shifts remains unexplored and challenging. To solve this problem, we propose Graph Invariant Learning (**GIL**) model capable of learning generalized graph representations under distribution shifts. Our proposed method can capture the invariant relationships between predictive graph structural information and labels in a mixture of latent environments through jointly optimizing three tailored modules. Specifically, we first design a GNN-based subgraph generator to identify invariant subgraphs. Then we use the variant subgraphs, i.e., complements of invariant subgraphs, to infer the latent environment labels. We further propose an invariant learning module to learn graph representations that can generalize to unknown test graphs. Theoretical justifications for our proposed method are also provided. Extensive experiments on both synthetic and real-world datasets demonstrate the superiority of our method against state-of-the-art baselines under distribution shifts for the graph classification task.

## 1   Introduction

Graph structured data is ubiquitous in the real world, e.g., social networks, biology networks, chemical molecules, etc. Graph representation learning, which encodes graphs into vectorized representations, has been the central topic in graph machine learning in the last decade. For example, graph neural networks (GNNs) [1–3] design end-to-end learning schemes to extract useful graph information and are shown to be successful in a variety of applications.

Despite the enormous success, the existing approaches for learning graph representations heavily rely on the I.I.D. assumption, i.e., the testing and training graph data are independently drawn from an identical distribution. However, distribution shifts of graph data widely exist in real-world scenarios and are usually inevitable due to the uncontrollable underlying data generation mechanism [4]. Most existing approaches fail to generalize to out-of-distribution (OOD) testing graph data. One critical bottleneck is that the existing methods ignore the *invariant* graph patterns and tend to rely on correlations that are *variant* for graphs from different environments. Therefore, it is of paramount significance to learn graph representations under distribution shifts and develop methods capable of out-of-distribution (OOD) generalization. Such studies are particularly critical for high-stake graph applications such as medical diagnosis [5], financial analysis [6], molecular prediction [7], etc.

In this work, we propose a brand new methodology to learn *invariant* graph representation under distribution shifts. Invariant learning, which aims to exploit the invariant relationships between

---
[*]Corresponding authors

36th Conference on Neural Information Processing Systems (NeurIPS 2022).

features and labels across different distributions while disregarding the variant spurious correlations, can provably achieve satisfactory OOD generalization under distribution shifts [8–10]. Though invariant learning has been studied for images and texts [8, 11], it remains largely unexplored in the literature of graph representation learning. However, invariant graph representation learning is non-trivial due to the following challenges. First, graph data usually comes from a mixture of latent environments without accurate environment labels, as shown in Figure 1. Since most invariant methods require multiple training environments with explicit environment labels, these existing methods cannot be directly applied to graphs. Second, the formation process of graphs is affected by the complex interaction of both invariant and variant patterns. How to identify the invariant patterns among latent environments is even more challenging. Last but not least, even after having obtained the environmental labels, how to design a theoretically grounded learning scheme to generate graph representations capable of OOD generalization remains largely unexplored.

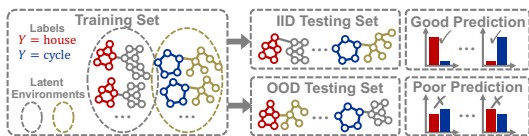

Figure 1: An example of distribution shifts under a mixture of latent environments, which leads to poor generalization.

To tackle these challenges, in this paper, we propose Graph Invariant Learning method (**GIL**) which is able to capture invariant graph patterns in a mixture of latent environments and capable of OOD generalization under distribution shifts. As shown in Figure 2, our proposed method can capture the invariant relationships between predictive graph structural information and labels in a mixture of latent environments through jointly optimizing three mutually promoting modules, with each module tackling one aforementioned challenge. Specifically, in the invariant subgraph identification module, we design a GNN-based subgraph generator to identify potentially invariant subgraphs from the complex interaction between invariant and variant patterns. Then, we use the variant subgraphs, i.e., the complement of invariant subgraphs, to infer environment labels by clustering these environment-discriminative features. The variant subgraphs capture variant correlations under different distributions and therefore contain informative features to infer environment labels. Lastly, in the invariant learning module, we propose to optimize the maximal invariant subgraph generator criterion given the identified invariant subgraphs and inferred environments to generate graph representations capable of OOD generalization under distribution shifts. We theoretically show that the OOD generalization problem on graphs can be formulated as finding a maximal invariant subgraph generator of our **GIL**, and further prove that our **GIL** satisfies permutation invariance. We conduct extensive experiments on both synthetic graph datasets and real graph benchmarks for the graph classification task. The results show that the representations learned from **GIL** achieve substantial performance gains on the unseen OOD testing graphs compared with various state-of-the-art baselines.

Our contributions are summarized as follows.

- We propose a novel Graph Invariant Learning method (**GIL**) to learn invariant and OOD generalized graph representations under distribution shifts. To the best of our knowledge, we are the first to study invariant learning for graph representation learning under a mixture of latent environments.

- Our proposed method can automatically infer the environment label of graphs from a mixture of latent environments without supervision.

- We propose maximal invariant subgraph generator criterion to learn graph representations capable of OOD generalization under distribution shifts.

- We theoretically show that finding a maximal invariant subgraph generator of **GIL** can solve the OOD generalization problem. Extensive empirical results demonstrate the effectiveness of **GIL** on various synthetic and benchmark datasets under distribution shifts.

## 2  Notations and Problem Formulation

**Notations.** Let $\mathbb{G}$ and $\mathbb{Y}$ be the graph and label space. We consider a graph dataset $\mathcal{G} = \{(G_i, Y_i)\}_{i=1}^{N}$ where $G_i \in \mathbb{G}$ and $Y_i \in \mathbb{Y}$. Following the OOD convention [11, 8, 9], we assume the dataset is collected from multiple training environments, i.e., $\mathcal{G} = \{\mathcal{G}^e\}_{e \in \mathrm{supp}(\mathcal{E}_{tr})}$, where $\mathcal{G}^e = \{(G_i^e, Y_i^e)\}_{i=1}^{N_e}$ denotes the dataset from environment $e$, $\mathrm{supp}(\mathcal{E}_{tr})$ is the support of the environmental variable in the training data. We use G and Y to denote the random variables of graph and label, and $G^e$ and $Y^e$ to

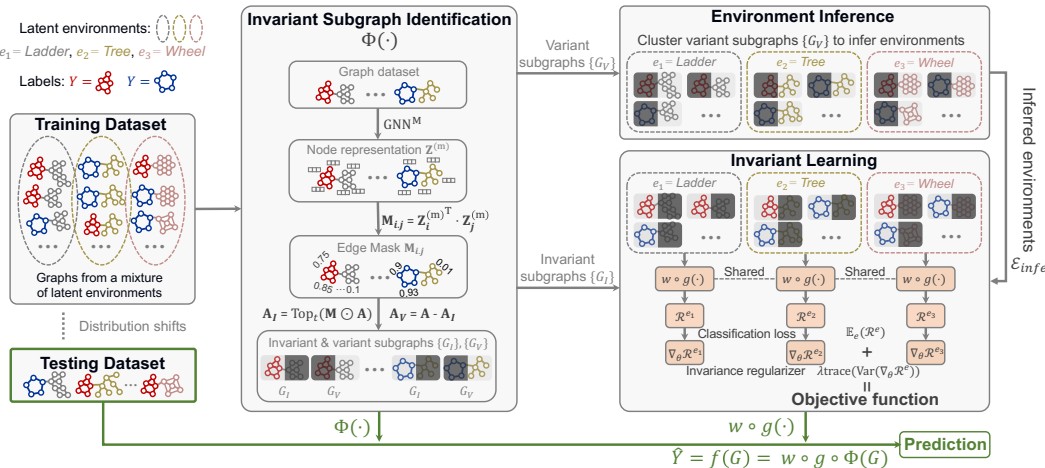

Figure 2: The framework of **GIL** model. Our proposed method jointly optimizes three modules: (1) In the invariant subgraph identification module, a GNN-based subgraph generator $\Phi(\cdot)$ identifies the invariant subgraph $G_I$ and the variant subgraph $G_V$. (2) The environment inference module uses the variant subgraphs $\{G_V\}$ to infer the latent environments by clustering the representations of $\{G_V\}$. (3) The invariant learning module jointly optimizes the invariant subgraph generator $\Phi(\cdot)$, the representation learning function $g(\cdot)$, and the classifier $w(\cdot)$. **Training stage** (shown by grey arrows): we back propagate with the objective function to update model parameters. **Testing stage** (shown by green arrows): we use the optimized model to make predictions.

specify random variables from environment $e$. The environment label for graphs is *unobserved* since it is prohibitively expensive to collect graph environment labels for most real scenarios.

**Problem Formulation.** We formulate the generalization under distribution shifts on graphs as:

**Problem 1.** *Let $\mathcal{E}$ denote the random variable on indices of **all** possible environments. Our goal is to find an optimal predictor $f^*(\cdot) : \mathbb{G} \to \mathbb{Y}$ that performs well on all environments:*

$$f^*(\cdot) = \arg\min_f \sup_{e \in \mathrm{supp}(\mathcal{E})} \mathcal{R}(f|e), \tag{1}$$

*where $\mathcal{R}(f|e) = \mathbb{E}^e_{\mathrm{G,Y}}[\ell(f(\mathrm{G}), \mathrm{Y})]$ is the risk of the predictor $f$ on the environment $e$, and $\ell(\cdot, \cdot) : \mathbb{Y} \times \mathbb{Y} \to \mathbb{R}$ denotes a loss function. We further decompose $f(\cdot) = w \circ h$, where $h(\cdot) : \mathbb{G} \to \mathbb{R}^d$ is the representation learning function, $d$ is the dimensionality, and $w(\cdot) : \mathbb{R}^d \to \mathbb{Y}$ is the classifier.*

Note that $\mathrm{supp}(\mathcal{E}_{tr}) \subset \mathrm{supp}(\mathcal{E})$. Besides, distribution shifts indicate that $P^e(\mathrm{G,Y}) \neq P^{e'}(\mathrm{G,Y}), e \in \mathrm{supp}(\mathcal{E}_{tr}), e' \in \mathrm{supp}(\mathcal{E}) \setminus \mathrm{supp}(\mathcal{E}_{tr})$, i.e., the joint distribution of the graph and the corresponding label is different for training and testing graph data.

Problem 1 is difficult to be solved since $\mathrm{supp}(\mathcal{E})$ is unobserved or latent [8, 9]. In addition, for most graph datasets, we do not have access to accurate environment labels or environment partitions. Therefore, we focus on jointly inferring the environments of the graph dataset $\mathcal{G}$ and achieving good OOD generalization performance under the inferred environments. The problem is formulated as:

**Problem 2.** *Given a graph dataset $\mathcal{G}$ collected from a mixture of latent environments but without environment labels, the task is to jointly infer graph environments $\mathcal{E}_{infer}$, i.e., $\mathcal{G} = \{\mathcal{G}^e\}_{e \in \mathrm{supp}(\mathcal{E}_{infer})}$, and learn a graph predictor $f^*(\cdot)$ in Problem 1 under the inferred environments $\mathcal{E}_{infer}$ to achieve good OOD generalization performance.*

## 3 Method

In this section, we introduce our proposed method in detail, whose framework is shown in Figure 2. We first present the invariant subgraph identification module. Then, we infer environment labels by clustering the variant subgraphs. Next, we introduce the maximal invariant subgraph generator criterion to generate graph representations which can generalize to test graphs under distribution shifts. Lastly, we provide some discussions of our proposed method.

## 3.1 Invariant Subgraph Identification

We assume that each input graph $G \in \mathcal{G}$ has an invariant subgraph $G_I \subset G$ so that its relationship with the label is invariant across different environments. We refer to the rest of the graph, i.e., the complement of $G_I$, as the variant subgraph and denote it as $G_V$. $G_V$ represents the graph part whose relationship with the label is variant across different environments, e.g., spurious correlations. Therefore, the model will have a better OOD generalization ability if it can identify the invariant subgraph and only uses structural information from $G_I$.

We denote a generator to obtain the invariant subgraph as $G_I = \Phi(G)$. Following the invariant learning literature [12], we make an assumption on $\Phi(G)$ as follows:

**Assumption 3.1.** *Given* G, *there exists an optimal invariant subgraph generator* $\Phi^*(G)$ *satisfying:*
*a.* Invariance property*:* $\forall e, e' \in \operatorname{supp}(\mathcal{E})$, $P^e(Y|\Phi^*(G)) = P^{e'}(Y|\Phi^*(G))$.
*b.* Sufficiency property*:* $Y = w^*(g^*(\Phi^*(G))) + \epsilon$, $\epsilon \perp G$, *where* $g^*(\cdot)$ *denotes a representation learning function,* $w^*$ *is the classifier,* $\perp$ *indicates statistical independence, and* $\epsilon$ *is random noise.*

The invariance assumption means that there exists a subgraph generator such that it can generate invariant subgraphs across different environments. The sufficiency assumption means that the generated invariant subgraphs should have sufficient predictive abilities in predicting the graph labels.

Under this assumption, we instantiate $\Phi(\cdot)$ with learnable parameters. Consider an input graph instance $G$ with $n$ nodes. The corresponding adjacency matrix is denoted as $\mathbf{A} = \{0, 1\}^{n \times n}$, where $\mathbf{A}_{i,j} = 1$ represents that there exists an edge between node $i$ and $j$, and $\mathbf{A}_{i,j} = 0$ otherwise. To split the input graph $G$ into $G_I$ and $G_V$, a common strategy is to use a binary mask matrix $\mathbf{M} = \{0, 1\}^{n \times n}$ on the adjacency matrix $\mathbf{A}$. However, directly optimizing a discrete matrix $\mathbf{M}$ is intractable as $G$ has exponentially many subgraph candidates [13]. Besides, learning $\mathbf{M}$ for each graph $G$ separately hinders the method from handling unseen test graphs [14]. Therefore, we adopt a shared learnable GNN (denoted as $\mathrm{GNN}^{\mathbf{M}}$) to generate a soft mask matrix $\mathbf{M} = \mathbb{R}^{n \times n}$ as follows:

$$\mathbf{M}_{i,j} = \mathbf{Z}_i^{(m)\top} \cdot \mathbf{Z}_j^{(m)}, \ \mathbf{Z}^{(m)} = \mathrm{GNN}^{\mathbf{M}}(G), \tag{2}$$

where $\mathbf{Z}^{(m)}$ is the node representation. Then, we obtain the invariant and variant subgraphs as:

$$\mathbf{A}_I = \mathrm{Top}_t \left( \mathbf{M} \odot \mathbf{A} \right), \mathbf{A}_V = \mathbf{A} - \mathbf{A}_I, \tag{3}$$

where $\mathbf{A}_I$ and $\mathbf{A}_V$ denotes the adjacency matrix of $G_I$ and $G_V$, respectively, $\odot$ means the element-wise matrix multiplication, and $\mathrm{Top}_t(\cdot)$ selects the top $t$-percentage of elements with the largest values. The parameters of $\mathrm{GNN}^{\mathbf{M}}$ are trained on all available graphs to generate the corresponding $G_I$ and $G_V$. Using the inductive learning ability of GNNs, it can also be used to unseen test graphs, as opposed to directly optimizing mask matrices.

## 3.2 Environment Inference

After obtaining the invariant and variant subgraphs, we can infer the environment label $\mathcal{E}_{infer}$. The intuition is that since the invariant subgraph captures invariant relationships between predictive graph structural information and labels, the variant subgraphs in turn capture variant correlations under different distributions, which are environment-discriminative features. Therefore, we can use the variant subgraphs to infer the latent environments. We adopt another GNN encoder, whose parameters are also shared among different graphs, to generate the representation of the variant subgraph $G_V$:

$$\mathbf{Z}_V = \mathrm{GNN}^{\mathbf{V}}(G_V), \ \mathbf{h}_V = \mathrm{READOUT}^{\mathbf{V}}(\mathbf{Z}_V), \tag{4}$$

where $\mathrm{READOUT}$ is a permutation-invariant readout function that aggregates node-level representation $\mathbf{Z}_V$ into graph-level representation $\mathbf{h}_V$. The representation of all the variant subgraphs is denoted as $\mathbf{H} = [\mathbf{h}_{V_1}, ..., \mathbf{h}_{V_N}]$. After obtaining $\mathbf{H}$, we use an off-the-shelf clustering algorithm to infer the environment label $\mathcal{E}_{infer}$. In this paper, we adopt k-means [15] as our clustering algorithm:

$$\mathcal{E}_{infer} = \mathrm{k\text{-}means}(\mathbf{H}). \tag{5}$$

Using $\mathcal{E}_{infer}$, we can partition the graph dataset into multiple training environments, i.e., $\mathcal{G} = \{\mathcal{G}^e\}_{e \in \operatorname{supp}(\mathcal{E}_{infer})}$. The environment inference module is purely unsupervised without needing any ground-truth environment labels.

### 3.3 Invariant Learning

After obtaining the inferred invariant subgraphs and environment labels, we propose the invariant learning module which can generate OOD generalized graph representations under distribution shifts.

Recall that both the invariant subgraph identification module and environment inference module heavily depend on the generator $\Phi$. Therefore, we aim to learn the optimal generator $\Phi^*$ in Assumption 3.1 by proposing and optimizing the **maximal invariant subgraph generator** criterion. First, following the invariant learning literature [9], we give the following definition.

**Definition 1.** *The **invariant subgraph generator set** $\mathcal{I}$ with respect to $\mathcal{E}$ is defined as:*

$$\mathcal{I}_{\mathcal{E}} = \{\Phi(\cdot) : P^e(\mathbf{Y}|\Phi(\mathbf{G})) = P^{e'}(\mathbf{Y}|\Phi(\mathbf{G})), e, e' \in \mathrm{supp}(\mathcal{E})\}. \tag{6}$$

Then, we show that the optimal generator $\Phi^*$ satisfies the following theorem.

**Theorem 3.2.** *A generator $\Phi(\mathbf{G})$ is the optimal generator that satisfies Assumption 3.1 if and only if it is the maximal invariant subgraph generator, i.e.,*

$$\Phi^* = \arg\max_{\Phi \in \mathcal{I}_{\mathcal{E}}} I\left(\mathbf{Y}; \Phi(\mathbf{G})\right), \tag{7}$$

*where $I(\cdot; \cdot)$ is the mutual information between the label and the generated subgraph.*

The proof is provided in Appendix. Eq. (7) provides us an objective function to optimize the subgraph generator. However, directly solving Eq. (7) for a non-linear $\Phi$ is difficult [9]. Following the invariant learning literature [9], we transform Eq. (7) into an invariance regularizer:

$$\mathbb{E}_{e \in \mathrm{supp}(\mathcal{E}_{infer})} \mathcal{R}^e(f(\mathbf{G}), \mathbf{Y}; \theta) + \lambda \mathrm{trace}(\mathrm{Var}_{\mathcal{E}_{infer}}(\nabla_\theta \mathcal{R}^e)), \tag{8}$$

where $f(\cdot) = w \circ g \circ \Phi$, $\mathcal{E}_{infer}$ is the infered environment label, and $\theta$ denotes all the learnable parameters. Recall that $g(\cdot)$ is the representation learning function of the invariant subgraphs and $w(\cdot)$ is the classifier. We instantiate $g$ as another GNN as follows:

$$\mathbf{Z}_I = \mathrm{GNN}^{\mathbf{I}}(G_I), \mathbf{h}_I = \mathrm{READOUT}^{\mathbf{I}}(\mathbf{Z}_I). \tag{9}$$

$\mathbf{Z}_I$ and $\mathbf{h}_I$ are the node-level and graph-level representations of invariant subgraph $G_I$, respectively. $w(\cdot)$ is instantiated as a multilayer perceptron followed by the softmax activation function. By optimizing Eq. (8), we can get our desired generator $\Phi$ and the subgraph representation learning function $g(\cdot)$, which collectively serve as our representation learning method $h(\cdot)$, i.e., $h = g \circ \Phi$.

### 3.4 Discussions

**Training Procedure.** We present the pseudocode of **GIL** in Appendix.

**Time Complexity.** The time complexity of our **GIL** is $O(|E| d + |V| d^2)$, where $|V|$ and $|E|$ denotes the number of nodes and edges, respectively, and $d$ is the dimensionality of the representations. Specifically, we adopt message-passing GNNs to instantiate our GNN components, which has a complexity of $O(|E| d + |V| d^2)$. Since we only need to generate mask for the existing edges in graphs, the time complexity of generating invariant and variant subgraphs and further obtaining their representations is $O(|E| d + |V| d^2)$. The time complexity of environment inference is $O(|\mathcal{B}||\mathcal{E}_{infer}|Td)$, where $|\mathcal{B}|$ is the batch size, $T$ is the number of iterations for the k-means algorithm, and $|\mathcal{E}_{infer}|$ denotes the number of inferred environments. The time complexity of the invariance regularizer is $O(|\mathcal{E}_{infer}|d^2)$, as the number of parameters for most GNNs is $O(d^2)$. Since $|\mathcal{B}|$, $|\mathcal{E}_{infer}|$, and $T$ are small constants, the overall time complexity of **GIL** is $O(|E| d + |V| d^2)$. In comparison, the time complexity of other GNN-based graph representation methods is also $O(|E| d + |V| d^2)$. Therefore, the time complexity of our proposed **GIL** is on par with the existing methods.

## 4 Theoretical Analysis

In this section, we theoretically analyze our **GIL** model by showing that the maximal invariant subgraph generator can achieve OOD optimal. The proofs are provided in Appendix.

**Theorem 4.1.** *Let $\Phi^*$ be the optimal invariant subgraph generator in Assumption 3.1 and denote the complement as $\mathbf{G}\backslash\Phi^*(\mathbf{G})$, i.e., the corresponding variant subgraph. Then, we can obtain the optimal predictor under distribution shifts, i.e., the solution to Problem 1, as follows:*

$$\arg\min_{w,g} w \circ g \circ \Phi^*(\mathbf{G}) = \arg\min_{f} \sup_{e \in \mathrm{supp}(\mathcal{E})} \mathcal{R}(f|e), \tag{10}$$

*if the following conditions hold: (1) $\Phi^*(\mathrm{G}) \perp \mathrm{G}\backslash\Phi^*(\mathrm{G})$; and (2) $\forall\Phi \in \mathcal{I}_{\mathcal{E}}$, $\exists\ e' \in \mathrm{supp}(\mathcal{E})$ such that $P^{e'}(\mathrm{G},\mathrm{Y}) = P^{e'}(\Phi(\mathrm{G}),\mathrm{Y})P^{e'}(\mathrm{G}\backslash\Phi(\mathrm{G}))$ and $P^{e'}(\Phi(\mathrm{G})) = P^e(\Phi(\mathrm{G}))$.*

The theorem shows that we can transform the OOD generalization problem into finding the optimal invariant subgraphs while maintaining the optimality.

We also prove that our **GIL** satisfies permutation invariance in Appendix.

## 5 Experiments

In this section, we evaluate the effectiveness of our **GIL** on both synthetic and real-world datasets.

### 5.1 Experimental Setup

**Datasets.** We adopt one synthetic dataset with controllable ground-truth environments and four real-world benchmark datasets for the graph classification task.
- **SP-Motif**: Following [13, 16], we generate a synthetic dataset where each graph consists of one variant subgraph and one invariant subgraph, i.e., motif. The variant subgraph includes Tree, Ladder, and Wheel (denoted by $V = 0, 1, 2$, respectively) and the invariant subgraph includes Cycle, House, and Crane (denoted by $I = 0, 1, 2$). The ground-truth label $Y$ only depends on the invariant subgraph $I$, which is sampled uniformly. The spurious correlation between $V$ and $Y$ is injected by controlling the variant subgraphs distribution as: $P(V) = r$ if $V = I$ and $P(V) = (1 - r)/2$ if $V \neq I$. Intuitively, $r$ controls the strength of the spurious correlation. We set $r$ to different values in the testing and training set to simulate the distribution shifts.
- **MNIST-75sp** [17]: The task is to classify each graph that is converted from an image in MNIST [18] into the corresponding handwritten digit. Distribution shifts exist on node features by adding random noises in the testing set.
- **Graph-SST2** [19]: Each graph is converted from a text sequence. Graphs are split into different sets based on average node degrees to create distribution shifts.
- **Open Graph Benchmark (OGB)** [20]: We consider two datasets, MOLSIDER and MOLHIV. The default split separates structurally different molecules with different scaffolds into different subsets.

**Baselines.** We compare our **GIL** with some representative state-of-the-art methods. The first group of these methods generates masks on graph structures to filter out important subgraphs using different GNNs, including Attention [2], Top-k Pool [21], SAGPool [22], and ASAP [23]. The second group is invariant learning methods, including standard ERM, GroupDRO [24], IRM [8], V-REx [25], DIR [16]. We also consider a recent interpretable graph learning method GSAT [26]. For a fair comparison, we use the same GNN backbone as **GIL** for the baselines.

**Optimization and Hyper-parameters.** The adopted GNNs and READOUT functions including $\mathrm{GNN}^{\mathbf{M}}$, $\mathrm{GNN}^{\mathbf{V}}$, $\mathrm{GNN}^{\mathbf{I}}$, $\mathrm{READOUT}^{\mathbf{V}}$, and $\mathrm{READOUT}^{\mathbf{I}}$ are listed in Appendix. The hyper-parameter $\lambda$ in Eq. (8) is chosen from $\{10^{-5}, 10^{-3}, 10^{-1}\}$. The number of clusters in Eq. (5) is chosen from $[2, 4]$. They are tuned on the validation set. We report the mean results and standard deviations of five runs. More details on the datasets, baselines and implementations are in Appendix.

### 5.2 Experiments on SP-Motif

**Settings.** To simulate different degrees of distribution shifts, we vary $r$ in both the training and testing datasets. For the training set, we select $r_{train}$ from $\{1/3, 0.5, 0.6, 0.7, 0.8, 0.9\}$. A larger $r_{train}$ indicates a higher spurious correlation between $Y$ and $\mathrm{G}_V$ in the training set, while $r_{train} = 1/3$ implies that the training set is balanced without any spurious correlation. For the testing set, we consider two settings: (1) $r_{test} = 1/3$, which simulates that the invariant subgraphs and variant subgraphs are randomly attached without spurious correlations; (2) $r_{test} = 0.2$, which indicates that the testing set has reversed spurious correlations and thus is more challenging.

**Quantitative Results.** The results are shown in Table 1. We have the following observations. Our proposed **GIL** model consistently and significantly outperforms the baselines and achieves the best performance on all settings. The results demonstrate that our proposed method can well handle graph distribution shifts and have a remarkable out-of-distribution generalization ability.

Table 1: The graph classification accuracy (%) on testing sets of the synthetic dataset SP-Motif. In each column, the boldfaced and the underlined score denotes the best and the second-best result, respectively. Numbers in the lower right corner denote standard deviations.

| $r_{train}$ | Scenario 1: $r_{test} = 1/3$ | | | | | | Scenario 2: $r_{test} = 0.2$ | | | | | |
| --- | --- | --- | --- | --- | --- | --- | --- | --- | --- | --- | --- | --- |
| | $r = 1/3$ | $r = 0.5$ | $r = 0.6$ | $r = 0.7$ | $r = 0.8$ | $r = 0.9$ | $r = 1/3$ | $r = 0.5$ | $r = 0.6$ | $r = 0.7$ | $r = 0.8$ | $r = 0.9$ |
| ERM | 53.60±3.79 | 51.24±4.13 | 47.04±7.01 | 38.80±3.72 | 37.84±3.01 | 37.44±2.15 | 48.48±4.53 | 41.72±4.81 | 36.92±6.93 | 35.72±8.33 | 28.80±3.91 | 19.60±1.66 |
| Attention | 54.31±3.98 | 53.24±3.56 | 42.52±6.20 | 35.20±1.05 | 34.48±1.18 | 33.88±1.01 | 44.04±4.33 | 31.64±0.67 | 25.72±5.34 | 24.80±4.06 | 23.20±3.60 | 18.04±2.88 |
| Top-k Pool | 54.68±2.71 | 53.12±5.58 | 44.56±4.57 | 37.44±2.04 | 35.24±2.28 | 34.28±4.11 | 45.68±5.16 | 34.20±4.34 | 31.00±2.89 | 30.64±3.59 | 29.16±2.18 | 27.56±3.91 |
| SAG Pool | 54.08±3.66 | 52.60±3.52 | 44.68±5.25 | 37.68±4.03 | 34.28±1.82 | 32.72±1.83 | 44.36±6.09 | 38.64±3.02 | 31.36±4.40 | 32.84±1.86 | 28.72±3.11 | 26.60±5.37 |
| ASAP | 54.00±4.21 | 51.92±3.81 | 45.12±1.98 | 36.28±0.86 | 34.24±2.02 | 34.40±3.15 | 49.88±4.90 | 34.52±4.35 | 27.00±2.61 | 27.20±2.53 | 27.96±3.89 | 22.88±4.33 |
| GroupDRO | 53.20±4.91 | 51.40±4.35 | 48.32±5.35 | 39.12±4.27 | 38.40±2.76 | 37.64±1.69 | 52.68±4.04 | 43.68±4.05 | 31.92±6.84 | 34.36±8.41 | 28.88±5.14 | 20.32±1.64 |
| IRM | 52.00±2.34 | 50.60±3.54 | 47.84±6.95 | 38.80±3.72 | 39.84±3.21 | 39.00±3.98 | 50.24±6.73 | 41.60±4.75 | 35.24±5.35 | 34.92±8.03 | 29.44±5.47 | 21.84±3.57 |
| V-REx | 53.16±3.25 | 46.04±6.11 | 45.36±3.66 | 40.24±3.86 | 39.48±3.00 | 39.12±3.48 | 50.56±2.83 | 37.16±6.24 | 34.52±3.00 | 29.72±4.58 | 27.32±3.18 | 24.04±6.08 |
| DIR | 52.96±5.06 | 52.08±1.93 | 50.12±2.76 | 49.84±2.46 | 45.20±1.11 | 41.24±4.73 | 50.68±5.20 | 49.96±1.75 | 45.44±6.00 | 40.56±2.36 | 39.92±4.53 | 32.52±4.59 |
| GSAT | 53.67±3.65 | 53.34±4.08 | 51.54±3.78 | 50.12±3.29 | 45.83±4.01 | 44.22±5.57 | 51.36±4.21 | 50.48±3.98 | 46.93±5.03 | 43.55±3.67 | 40.35±4.21 | 33.87±5.19 |
| GIL | 55.44±3.11 | 54.56±3.02 | 53.60±4.82 | 53.12±2.18 | 51.24±3.88 | 46.04±3.51 | 54.80±3.93 | 52.48±4.41 | 50.08±5.47 | 47.44±2.87 | 46.36±3.80 | 35.80±5.03 |

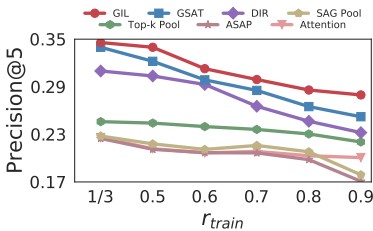

Figure 3: The results of discovering the ground-truth invariant subgraphs on SP-Motif ($r_{test} = 0.2$).

Figure 4: The test accuracy and the performance of environment inference over different training periods.

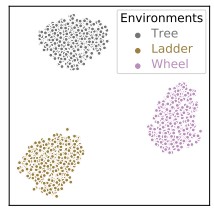

Figure 5: The environment inference results when training is finished.

As $r_{train}$ grows larger, the performance of all the methods tends to decrease since there exists a larger degree of distribution shift. Nevertheless, our proposed method is able to maintain the most relatively stable performance. In fact, the performance gap between **GIL** and the baselines becomes more significant as the degree of distribution shift increases. For example, when $r_{test} = 1/3$, the accuracy of all baselines drops by more than 7% when $r_{train}$ changes from 0.5 to 0.8, indicating their poor OOD generalization ability. In contrast, our method only has 3% performance drop.

When the degree of distribution shift is relatively small, GNNs with different pooling methods to generate subgraphs generally report better results. On the other hand, when the degree of distribution shift is large, invariant baselines show more stable performance. Among them, DIR, which is a recently proposed invariant method specifically designed for graphs, is one competitive baseline. Nevertheless, our proposed method outperforms DIR by more than 3% in terms of the classification accuracy in most cases. GSAT achieves promising gains over the other baselines, but our **GIL** still performs better than GSAT. When $r_{test} = r_{train} = 1/3$, i.e., no distribution shifts, our proposed method also achieves the best results, indicating that learning invariant subgraphs is also beneficial.

**Analysis.** To analyze whether our proposed method can accurately capture the invariant subgraph, we compare **GIL** with baselines that also output subgraphs using the ground-truth invariant subgraphs. The evaluation metric is Precision@5. We report the results in Figure 3. The results show that **GIL** has a clear advantage in discovering invariant subgraphs under latent environments, while the other baselines cannot handle distribution shifts well.

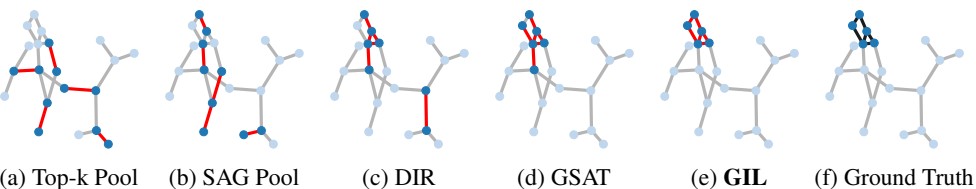

(a) Top-k Pool    (b) SAG Pool    (c) DIR    (d) GSAT    (e) **GIL**    (f) Ground Truth

Figure 6: Visualizations of the learned invariant subgraph for a showcase from the testing set of SP-Motif. In Figures (a)-(e), the red lines indicate the learned invariant subgraph, and the ground-truth is shown by the black lines in Figure (f).

Table 2: The graph classification results (%) on testing sets of the real-world datasets. We report the accuracy for MNIST-75sp and Graph-SST2, ROC-AUC for MOLSIDER and MOLHIV.

|  | MNIST-75sp | Graph-SST2 | MOLSIDER | MOLHIV |
|---|---|---|---|---|
| ERM | $14.94_{\pm 3.27}$ | $81.44_{\pm 0.59}$ | $57.57_{\pm 1.56}$ | $76.20_{\pm 1.14}$ |
| Attention | $16.44_{\pm 3.78}$ | $81.57_{\pm 0.71}$ | $56.99_{\pm 0.54}$ | $75.84_{\pm 1.33}$ |
| Top-k Pool | $15.02_{\pm 3.08}$ | $79.78_{\pm 1.35}$ | $60.63_{\pm 1.52}$ | $73.01_{\pm 1.65}$ |
| SAG Pool | $19.34_{\pm 1.73}$ | $80.24_{\pm 1.72}$ | $61.29_{\pm 1.31}$ | $73.26_{\pm 0.84}$ |
| ASAP | $15.14_{\pm 3.58}$ | $81.57_{\pm 0.84}$ | $55.77_{\pm 1.34}$ | $73.81_{\pm 1.17}$ |
| GroupDRO | $15.72_{\pm 4.35}$ | $81.29_{\pm 1.44}$ | $56.31_{\pm 1.15}$ | $75.44_{\pm 2.70}$ |
| IRM | $18.74_{\pm 2.43}$ | $81.01_{\pm 1.13}$ | $57.10_{\pm 0.92}$ | $74.46_{\pm 2.74}$ |
| V-REx | $18.40_{\pm 1.12}$ | $81.76_{\pm 0.08}$ | $57.76_{\pm 0.78}$ | $75.62_{\pm 0.79}$ |
| DIR | $17.38_{\pm 3.52}$ | $83.29_{\pm 0.53}$ | $57.74_{\pm 1.63}$ | $77.05_{\pm 0.57}$ |
| GSAT | $20.12_{\pm 1.35}$ | $82.95_{\pm 0.58}$ | $60.82_{\pm 1.36}$ | $76.47_{\pm 1.53}$ |
| **GIL** | $\mathbf{21.94_{\pm 0.38}}$ | $\mathbf{83.44_{\pm 0.37}}$ | $\mathbf{63.50_{\pm 0.57}}$ | $\mathbf{79.08_{\pm 0.54}}$ |

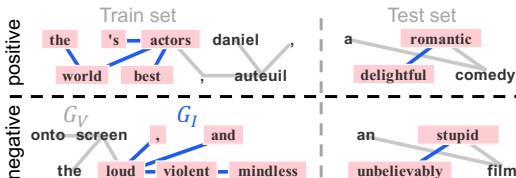

Figure 7: Four showcases of sentences with positive/negative sentiments of train/test sets on Graph-SST2 learned by our **GIL**. Blue edges indicate the learned invariant subgraphs, while the others are variant subgraphs.

Besides the quantitative evaluation, we plot a showcase from the testing set of SP-Motif ($r_{train} = 0.8$ and $r_{test} = 0.2$) in Figure 6. The figure shows that the learned invariant subgraph of our method is more accurate than baselines.

## 5.3 Experiments on Real-world Graphs

We further evaluate the effectiveness of our method on real-world graph datasets. The experimental results are presented in Table 2. Our **GIL** achieves the best performance on all four datasets, indicating that **GIL** can well handle distribution shifts on real-world graphs. For example, **GIL** increases the classification accuracy by 1.8% on MNIST-75sp and ROC-AUC by 2.0% on MOLHIV against the strongest baselines respectively. On MOLHIV, the results of most baselines are worse than ERM, indicating that they fail to achieve OOD generalization in this dataset. Besides, different datasets have different distribution shifts, e.g., Graph-SST2 has different node degrees, the distribution shift of MNIST-75sp is on node features, and OGB is split based on scaffold. Therefore, the results show that our proposed method can well handle diverse types of distribution shifts in real graph datasets.

For MOLHIV, besides adopting GIN [3] as backbone (shown in Table 2), our method is also compatible with the other popular GNNs. We try using HIG and CIN [27] (Rank #2 and #8 on

Table 3: The test results with different backbones.

| CIN (Rank #8) | **GIL** (CIN Backbone) | HIG (Rank #2) | PAS+FPs (Rank #1) | **GIL** (HIG Backbone) |
|---|---|---|---|---|
| $80.94_{\pm 0.57}$ | $\mathbf{81.15_{\pm 0.46}}$ | $84.03_{\pm 0.21}$ | $84.20_{\pm 0.15}$ | $\mathbf{84.23_{\pm 0.25}}$ |

the MOLHIV leaderboard[2]) as the backbone since these models are orthogonal to ours. Table 3 shows that our **GIL** can consistently improve these models.

In addition, we present some showcases of the learned invariant subgraph of the proposed **GIL** on both the train and test set of Graph-SST2. This dataset consists of sentences with positive/negative sentiments and is more understandable for humans. Figure 7 shows that our method can learn invariant subgraphs by consistently focusing on the positive/negative words that are salient for sentiments and distinguishing invariant/variant parts under distribution shifts. For example, the subgraph "The world's best actors" identified by **GIL** has a predictive and invariant relationship with the positive sentiment label, while the subgraph "daniel auteuil" may reflect variant sentiments in different sentences. These results validate: (1) the invariant and variant subgraphs widely exist in real-world datasets, and (2) our **GIL** can well identify invariant subgraphs under distribution shifts and further make predictions with high accuracy based on the learned invariant subgraphs.

## 5.4 Analysis of Environment Inference

In our proposed model, all components are jointly optimized. To show that the environment inference module and invariant learning module can mutually enhance each other, we record the test accuracy and the Silhouette score [28], which is a commonly used evaluation metric for clustering, as the model is trained. The results on SP-Motif ($r_{train} = 0.8, r_{test} = 1/3$) are shown in Figure 4. We can observe that the test accuracy and the clustering performance improve synchronously over training. A plausible reason is that, as the training stage progresses, the invariant subgraph generator is optimized so that it can generate more informative invariant subgraphs and therefore improve the performance on the testing set. On the other hand, accurately discovering invariant subgraphs can also promote

---

[2]https://ogb.stanford.edu/docs/leader_graphprop/#ogbg-molhiv

identifying variant subgraphs, which capture the environment-discriminate features and better infer the latent environments. To verify that **GIL** can infer the environments accurately, we use t-SNE [29] to plot the discovered environments on a 2D-plane when the optimization is finished. Figure 5 shows that the variant subgraphs perfectly capture the environment-discriminate features. Notice that **GIL** achieves such results without needing any ground-truth environment label.

### 5.5 Hyper-parameter Sensitivity

We investigate the sensitivity of hyper-parameters of our method, including the number of environments $|\mathcal{E}_{infer}|$, the invariance regularizer coefficient $\lambda$, and the size of the invariant subgraph mask $t$ in Eq. (3). For simplicity, we only report the results on SP-Motif ($r_{train} = 0.8$ and $r_{test} = 1/3$) and MNIST-75sp in Figure 8, while the results on other datasets show similar patterns.

First, the number of environments has a moderate impact on the model performance. For SP-Motif, the performance reaches a peak when $|\mathcal{E}_{infer}| = 3$, showing that **GIL** achieves the best result when the number of environments matches the ground truth. For MNIST-75sp, the best number of environments is $|\mathcal{E}_{infer}| = 2$. A plausible reason is that a large number of environments will bring difficulty in inferring the latent environment, leading to sub-optimal performance. Second, the coefficient $\lambda$ also has a slight influence on the performance, indicating that we need to properly balance the classification loss and the invariance regularizer term. Finally, a proper value of the mask size $t$ is important. A very large $t$ will result in too many edges in the invariant subgraph and bring in variant structures, while a small $t$ may let the invariant subgraph become too small to capture enough structural information. Although an appropriate choice of hyper-parameters can further improve the performance, our method is not very sensitive to hyper-parameters. Figure 8 shows that **GIL** can outperform the best baselines with a wide range of hyper-parameters choices.

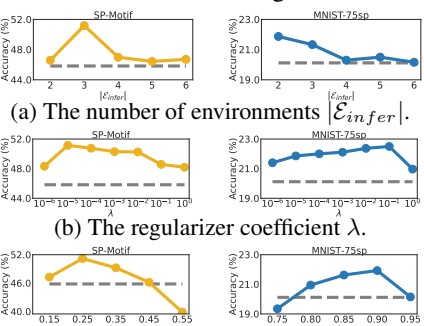

(a) The number of environments $|\mathcal{E}_{infer}|$.

(b) The regularizer coefficient $\lambda$.

(c) The invariant subgraph mask size $t$.

Figure 8: The impact of different hyper-parameters. Yellow and blue lines denote the results of **GIL** and grey dashed lines are the best results of all baselines.

## 6   Related Works

**Graph neural networks.** Recently, graph neural networks (GNNs) have shown enormous success in graph representation learning [1–3], demonstrating their strength in various tasks [30–36]. GNNs generally adopt a neighborhood aggregation (message passing) paradigm, i.e., the representations of nodes are iteratively updated by aggregating representations of their neighbors. The representation of the whole graph is summarized on node representations through the readout function (i.e., pooling) [3, 22]. However, most existing GNN models do not consider the out-of-distribution generalization ability [37] so that their performances can drop substantially on testing graphs with distribution shifts.

**Generalization of GNNs.** Early works [38–41] for analyzing the generalization ability of GNNs do not consider distribution shifts [37, 42, 43]. More recently, the generalization ability of GNNs under distribution shifts starts to receive research attention [44, 16, 45, 46]. [47] find that encoding task-specific non-linearities in the architecture or features can improve GNNs in extrapolating graph algorithmic tasks. [17, 48, 49] try to encourage GNNs to perform well on testing graphs with different sizes. Some works [50, 51] are proposed to deal with node-level tasks. EERM [52] studies the OOD generalization in node classification. However, little attention has been paid to learning graph-level representations under distribution shifts from the invariant learning perspective. One exception is the work DIR [16], which conducts interventions on graphs to create interventional distributions. However, performing causal intervention relies on strong assumptions [53] that could be violated and expensive to satisfy in practice [54]. GSAT [26] applies graph information bottleneck criteria for generalization, but its goal is mainly to build inherently interpretable GNNs.

**Invariant Learning.** Invariant learning aims to exploit the invariant relationships between features and labels across distribution shifts, while filtering out the variant spurious correlations. Backed by causal theory, invariant learning model can lead to OOD optimal models under some assumptions [11,

8, 9]. However, most existing methods heavily rely on multiple environments that have to be explicitly provided in the training dataset. Such annotation is not only prohibitively expensive for graphs, but also inherently problematic as the environment split could be inaccurate, rendering these invariant learning methods inapplicable. A few works study OOD generalization on latent environments in computer vision [55, 56] or raw feature data [57], which cannot be directly applied to graphs. In summary, how to learn invariant graph representations without explicit environment label under distribution shits remains largely unexplored in the literature.

## 7   Conclusions

In this paper, we propose the graph invariant learning (**GIL**) model to tackle the problem of learning invariant graph representations under distribution shifts. Three tailored modules are jointly optimized to encourage the graph representations to capture the invariant relationships between predictive graph structural information and labels. Theoretical analysis and extensive experiments on both synthetic and real-world datasets demonstrate the superiority of **GIL**.

## Acknowledgements

This work was supported in part by the National Key Research and Development Program of China No. 2020AAA0106300 and National Natural Science Foundation of China (No. 62250008, 62222209, 62102222, 62206149), China National Postdoctoral Program for Innovative Talents No. BX20220185, China Postdoctoral Science Foundation No. 2022M711813.

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
