# Learning Invariant Graph Representations for Out-of-Distribution Generalization
## (Appendix)

**Haoyang Li,  Ziwei Zhang,  Xin Wang,  Wenwu Zhu**
Tsinghua University
lihy18@mails.tsinghua.edu.cn, {zwzhang,xin_wang,wwzhu}@tsinghua.edu.cn

## A  Notations

We summarize the key notations and the corresponding descriptions in Table 1.

Table 1: Notations.

| Notation | Description |
|---|---|
| $N$ | The number of graphs in the dataset |
| $\mathbb{G}, \mathbb{Y}$ | The graph space and the label space |
| $\mathbf{G}, \mathbf{Y}$ | A random variable of graph and label |
| $G, Y$ | An instance of graph and label |
| $G_I = \Phi(G)$ | An instance of the invariant subgraph and the invariant subgraph generator |
| $\Phi^*$ | The optimal invariant subgraph generator |
| $G_V = G \backslash G_I$ | An instance of the variant subgraph |
| $\mathbf{A}_I / \mathbf{A}_V$ | The adjacency matrix of the invariant/variant subgraph |
| $\mathbf{Z}_I / \mathbf{Z}_V$ | The node-level representations of the invariant/variant subgraph |
| $\mathbf{h}_I / \mathbf{h}_V$ | The graph-level representations of the invariant/variant subgraph |
| $\mathcal{E} / \mathcal{E}_{tr}$ | A random variable on indices of all/training environments |
| $\mathcal{E}_{infer}$ | A random variable on indices of the inferred environments |
| $e$ | An instance of environment |
| $f$ | The predictor from $\mathbb{G}$ to $\mathbb{Y}$ |
| $w$ | The classifier from $\mathbb{R}^d$ to $\mathbb{Y}$ |
| $h$ | The representation learning function from $\mathbb{G}$ to $\mathbb{R}^d$ |
| $g$ | The representation learning function for invariant subgraphs |
| $\mathcal{I}_\mathcal{E}$ | The invariant subgraph generator set with respect to $\mathcal{E}$ |
| $\ell$ | The loss function |

## B  Training Procedure

To show the training procedure, we present the pseudocode of **GIL** in Algorithm 1.

## C  Explanations of Assumption

In the main paper, we denote the invariant subgraph generator as $\Phi(\cdot)$ and make the following assumption on $\Phi(\mathbf{G})$:

**Assumption 3.1.** *Given* $\mathbf{G}$, *there exists an optimal invariant subgraph generator* $\Phi^*(\mathbf{G})$ *satisfying:*
*a. Invariance property:* $\forall e, e' \in \mathrm{supp}(\mathcal{E}), P^e(\mathbf{Y}|\Phi^*(\mathbf{G})) = P^{e'}(\mathbf{Y}|\Phi^*(\mathbf{G}))$.
*b. Sufficiency property:* $\mathbf{Y} = w^*(g^*(\Phi^*(\mathbf{G}))) + \epsilon, \ \epsilon \perp \mathbf{G}$, *where* $g^*(\cdot)$ *denotes a representation learning function,* $w^*$ *is the classifier,* $\perp$ *indicates statistical independence, and* $\epsilon$ *is random noise.*

36th Conference on Neural Information Processing Systems (NeurIPS 2022).

---

**Algorithm 1** The training procedure of our **GIL**.

---

**Input:** Graph dataset $\mathcal{G} = \{(G_i, Y_i)\}_{i=1}^N$
**Output:** An optimized predictor $f(\cdot) : \mathbb{G} \rightarrow \mathbb{Y}$

1: **for** sampled minibatch $\mathcal{B}$ of graph dataset $\mathcal{G}$ **do**
2:     **for** each graph instance $(G, Y) \in \mathcal{B}$ **do**
3:         Generate the mask matrix $\mathbf{M}$ with the shared learnable $\text{GNN}^{\mathbf{M}}$ by Eq. (2).
4:         Obtain the invariant subgraph $G_I$ and the variant subgraph $G_V$ by Eq. (3).
5:         Generate representations $\mathbf{h}_V$ of variant subgraph $G_V$ by Eq. (4).
6:         Generate representations $\mathbf{h}_I$ of invariant subgraph $G_I$ by Eq. (9).
7:     **end for**
8:     Infer environments $\mathcal{E}_{infer}$ with clustering representations of variant subgraphs $\mathbf{H}$ by Eq. (5).
9:     Calculate the objective function by Eq. (8).
10:    Update model parameters using back propagation.
11: **end for**

---

The invariance assumption means that there exists a subgraph generator such that it can generate invariant subgraphs across different environments. The sufficiency assumption means that the generated invariant subgraphs should have sufficient predictive abilities in predicting the graph labels. We make this assumption following the literature, e.g, [1–4].

To better illustrate this assumption is commonly satisfied, we provide real-world showcases in Figure 1. For molecule graphs from [3], invariant subgraph $G_I$ (indicated by blue edges) represents "hydrophilic R-OH group"/"non-polar repeated ring structures", whose relationship with the label solubility/anti-solubility is truly predictive and invariant across different environments. And variant subgraph $G_V$ denotes carbon structure or scaffold [4]. For superpixel graphs from [2], $G_I$ and $G_V$ represent the edges corresponding to the digit itself and other edges from the background, respectively.

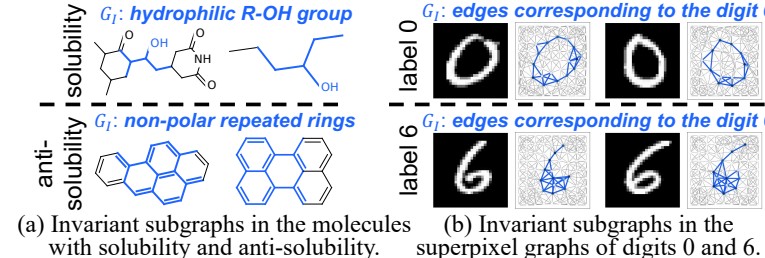

(a) Invariant subgraphs in the molecules    (b) Invariant subgraphs in the
with solubility and anti-solubility.      superpixel graphs of digits 0 and 6.

Figure 1: Graph examples in real-world scenarios that include invariant $G_I$ (blue edges) and variant $G_V$ (gray edges) subgraphs.

## D   Proofs

In this section, we provide the proofs of Theorem 3.2 and 4.1. We also theoretically analyze that our **GIL** satisfies permutation invariance in Section D.3.

### D.1   Proof of Theorem 3.2

**Theorem 3.2.** *A generator $\Phi(\mathbf{G})$ is the optimal generator that satisfies Assumption 3.1 if and only if it is the maximal invariant subgraph generator, i.e.,*

$$\Phi^* = \arg \max_{\Phi \in \mathcal{I}_{\mathcal{E}}} I\left(\mathbf{Y}; \Phi(\mathbf{G})\right), \tag{1}$$

*where $I(\cdot; \cdot)$ is the mutual information between the label and the generated subgraph.*

*Proof.* Denote $\hat{\Phi} = \arg\max_{\Phi \in \mathcal{I}_{\mathcal{E}}} I\left(Y; \Phi(G)\right)$. From the invariance property of Assumption 3.1, $\Phi^* \in \mathcal{I}_{\mathcal{E}}$. Therefore, we prove the theorem by showing that $I(Y; \hat{\Phi}(G)) \leq I(Y; \Phi^*(G))$ and consequently, $\hat{\Phi} = \Phi^*$.

To show the inequality, we use the functional representation lemma [5], which states that for any random variables $X_1$ and $X_2$, there exists a random variable $X_3$ independent of $X_1$ such that $X_2$ can be represented as a function of $X_1$ and $X_3$. So for $\Phi^*(G)$ and $\hat{\Phi}(G)$, there exists $\Phi'(G)$ satisfying that $\Phi'(G) \perp \Phi^*(G)$ and $\hat{\Phi}(G) = \gamma\left(\Phi^*(G), \Phi'(G)\right)$, where $\gamma(\cdot)$ is a function. Then, we can derive that:

$$
\begin{aligned}
I(Y; \hat{\Phi}(G)) &= I\left(Y; \gamma\left(\Phi^*(G), \Phi'(G)\right)\right) \\
&\leq I\left(Y; \Phi^*(G), \Phi'(G)\right) \\
&= I\left(w^*(g^*(\Phi^*(G))); \Phi^*(G), \Phi'(G)\right) \\
&= I\left(w^*(g^*(\Phi^*(G))); \Phi^*(G)\right) = I\left(Y; \Phi^*(G)\right),
\end{aligned}
\tag{2}
$$

which finishes the proof. $\qquad\square$

### D.2   Proof of Theorem 4.1

**Theorem 4.1.** *Let $\Phi^*$ be the optimal invariant subgraph generator in Assumption 3.1 and denote the complement as $G \backslash \Phi^*(G)$, i.e., the corresponding variant subgraph. Then, we can obtain the optimal predictor under distribution shifts, i.e., the solution to Problem 1, as follows:*

$$
\arg\min_{w,g} w \circ g \circ \Phi^*(G) = \arg\min_{f} \sup_{e \in \mathrm{supp}(\mathcal{E})} \mathcal{R}(f|e),
\tag{3}
$$

*if the following conditions hold: (1) $\Phi^*(G) \perp G \backslash \Phi^*(G)$; and (2) $\forall \Phi \in \mathcal{I}_{\mathcal{E}}, \exists\, e' \in \mathrm{supp}(\mathcal{E})$ such that $P^{e'}(G, Y) = P^{e'}(\Phi(G), Y) P^{e'}(G \backslash \Phi(G))$ and $P^{e'}(\Phi(G)) = P^e(\Phi(G))$.*

*Proof.* Denote the function to obtain the complement of invariant subgraph as $\Psi(G) = G \backslash \Phi(G)$ and $\Psi^*(G) = G \backslash \Phi^*(G)$. By assumption, $\Phi^*(G) \perp \Psi^*(G)$. Further denote $\hat{f} = \arg\min_{w,g} w \circ g \circ \Phi^*(G)$. By Assumption 3.1, we have

$$
\hat{f}(G) = w^* \circ g^* \circ \Phi^*(G).
\tag{4}
$$

To show that $\hat{f}$ is $f^*$, our proof strategy is to show that $\forall e \in \mathrm{supp}(\mathcal{E})$, for any possible $f$, $\mathcal{R}(\hat{f}|e) \leq \mathcal{R}(f|e')$ and therefore $\sup_{e \in \mathrm{supp}(\mathcal{E})} \mathcal{R}(\hat{f}|e) \leq \sup_{e \in \mathrm{supp}(\mathcal{E})} \mathcal{R}(f|e)$.

To show the inequality, we have:

$$
\mathcal{R}(\hat{f}|e) = \mathbb{E}^e_{G,Y}[\ell(\hat{f}(G), Y)] = \sum_{G,Y} P^e(G, Y)\ell(\hat{f}(G), Y)
\tag{5}
$$

$$
= \sum_{\Psi^*(G)} P^e(\Psi^*(G)) \sum_{\Phi^*(G),Y} P^e(\Phi^*(G), Y)\ell\left(w^*(g^*(\Phi^*(G))), Y\right)
\tag{6}
$$

$$
= \sum_{\Phi^*(G),Y} P^e(\Phi^*(G), Y)\ell(w^*(g^*(\Phi^*(G))), Y)
\tag{7}
$$

$$
\leq \sum_{\Phi(G),Y} P^e(\Phi(G), Y)\ell(w(g(\Phi(G))), Y)
\tag{8}
$$

$$
= \sum_{\Psi(G)} P^{e'}(\Psi(G)) \sum_{\Phi(G),Y} P^e(\Phi(G), Y)\ell(w(g(\Phi(G))), Y)
\tag{9}
$$

$$
= \sum_{\Psi(G)} \sum_{\Phi(G),Y} P^{e'}(\Phi(G), Y) P^{e'}(\Psi(G))\ell(w(g(\Phi(G))), Y)
\tag{10}
$$

$$
= \sum_{G,Y} P^{e'}(G, Y)\ell(f(G), Y) = \mathbb{E}^{e'}_{G,Y}[\ell(f(G), Y)] = \mathcal{R}(f|e').
\tag{11}
$$

$\qquad\square$

### D.3 Proof of Permutation-invariance of GIL

**Theorem D.3.** *Our proposed **GIL** model is permutation-invariant if* $\text{GNN}^{\mathbf{M}}$ *and* $\text{GNN}^{\mathbf{I}}$ *are permutation-equivariant and* $\text{READOUT}^{\mathbf{I}}$ *is permutation-invariant.*

*Proof.* The theorem is straight-forward from the compositionality of permutation-equivariant and invariant functions. Concretely, recall our learned model $f = w \circ g \circ \Phi$ is formulated as:

$$
\begin{aligned}
\Phi : \mathbf{A}_I &= \text{Top}_t \left( \mathbf{M} \odot \mathbf{A} \right), \mathbf{M}_{i,j} = \mathbf{Z}_i^{(m)^\top} \cdot \mathbf{Z}_j^{(m)}, \mathbf{Z}^{(m)} = \text{GNN}^{\mathbf{M}}(G). \\
g : \mathbf{h}_I &= \text{READOUT}^{\mathbf{I}}(\mathbf{Z}_I), \mathbf{Z}_I = \text{GNN}^{\mathbf{I}}(G_I) \\
w : \hat{Y} &= \text{MLP}(\mathbf{h}_I).
\end{aligned}
\tag{12}
$$

Consider $G' = \pi(G)$, where $\pi$ is a permutation of nodes. We denote all variables for $G'$ with a prime symbol in the top right corner. Since $\text{GNN}^{\mathbf{M}}$ is permutation-equivariant, we have:

$$
\mathbf{Z}^{(m)\prime} = \text{GNN}^{\mathbf{M}}(G') = \text{GNN}^{\mathbf{M}}(\pi(G)) = \pi(\text{GNN}^{\mathbf{M}}(G)) = \pi(\mathbf{Z}^{(m)}).
\tag{13}
$$

Since the inner product matrix and $\text{Top}_t(\cdot)$ are also equivariant with respect to permutations, we easily have:

$$
\begin{aligned}
\mathbf{M}'_{i,j} &= \mathbf{Z}_i^{(m)\prime^\top} \cdot \mathbf{Z}_j^{(m)\prime} = \pi(\mathbf{Z}_i^{(m)})^\top \cdot \pi(\mathbf{Z}_j^{(m)}) = \pi(\mathbf{Z}_i^{(m)^\top} \cdot \mathbf{Z}_j^{(m)}) = \pi(\mathbf{M}_{i,j}) \\
\mathbf{A}'_I &= \text{Top}_t(\mathbf{M}' \odot \mathbf{A}') = \text{Top}_t(\pi(\mathbf{M}) \odot \pi(\mathbf{A})) = \pi(\text{Top}_t(\mathbf{M} \odot \mathbf{A})) = \pi(\mathbf{A}_I)
\end{aligned}
\tag{14}
$$

Since $\text{GNN}^{\mathbf{I}}$ are permutation-equivariant and $\text{READOUT}^{\mathbf{I}}$ is permutation-invariant, we have:

$$
\begin{aligned}
\mathbf{Z}'_I &= \text{GNN}^{\mathbf{I}}(G'_I) = \text{GNN}^{\mathbf{I}}(\pi(G_I)) = \pi(\text{GNN}^{\mathbf{I}}(G_I)) = \pi(\mathbf{Z}_I) \\
\mathbf{h}'_I &= \text{READOUT}^{\mathbf{I}}(\mathbf{Z}'_I) = \text{READOUT}^{\mathbf{I}}(\pi(\mathbf{Z}_I)) = \mathbf{h}_I.
\end{aligned}
\tag{15}
$$

Therefore, we have $\hat{Y}' = \text{MLP}(\mathbf{h}'_I) = \text{MLP}(\mathbf{h}_I) = \hat{Y}$, i.e., $f$ is permutation-invariant. $\square$

The theorem shows that while learning invariant graph representations under distribution shifts, our proposed method naturally holds permutation-invariance as other GNNs.

## E  Experimental Details

### E.1  Datasets

Table 2: The statistics of the datasets. #Graphs(Train/Val/Test) is the number of graphs in the training/validation/testing set of the dataset. #Classes is the number of classes. Average #Nodes/#Edges are the average number of nodes and edges in the graph of the dataset, respectively.

|  | SP-Motif | MNIST-75sp | Graph-SST2 | OGBG-MOLSIDER | OGBG-MOLHIV |
|---|---|---|---|---|---|
| #Graphs(Train/Val/Test) | 1,500/500/500 | 5,000/1,000/1,000 | 28,327/3,147/12,305 | 1,141/143/143 | 32,901/4,113/4,113 |
| #Classes | 3 | 10 | 2 | 2 | 2 |
| Avg #nodes | 26.7 | 66.8 | 13.7 | 33.6 | 25.5 |
| Avg #edges | 36.7 | 600.2 | 25.3 | 35.4 | 27.5 |

We adopt one synthetic dataset with controllable ground-truth environments and four real-world benchmark datasets for the graph classification task. The statistics of these datasets are provided in Table 2.

- **SP-Motif**: Following [6, 2], we generate a synthetic dataset where each graph consists of one variant subgraph and one invariant subgraph, i.e., motif. The variant subgraph includes Tree, Ladder, and Wheel (denoted by $V = 0, 1, 2$, respectively) and the invariant subgraph includes Cycle, House, and Crane (denoted by $I = 0, 1, 2$). The ground-truth label $Y$ only depends on the invariant subgraph $I$, which is sampled uniformly. Besides, we inject a spurious correlation between $V$ and $Y$ by controlling the variant subgraphs distribution as:

$$
P(V) = \begin{cases} r, & \text{if } V = I \\ (1-r)/2, & \text{if } V \neq I \end{cases}.
\tag{16}
$$

Intuitively, $r$ controls the strength of the spurious correlation between $V$ and $Y$. We set $r$ to different values in the testing and training set to simulate the distribution shifts, i.e., $r_{train} \in \{1/3, 0.5, 0.6, 0.7, 0.8, 0.9\}$ and $r_{test} \in \{1/3, 0.2\}$. For this dataset, we adopt random node features and constant edge weights.

- **MNIST-75sp** [7]: Each graph is converted from an image in MNIST [8] using superpixels [9]. We sample 7,000 images to build our dataset. The nodes are superpixels, and the edges are calculated by the spatial distance between nodes. The node features are set as the super-pixel coordinates and intensity. The task is to classify each graph into the corresponding handwritten digit labeled from $0$ to $9$. To simulate distribution shifts with respect to graph features, we follow [7] and generate testing graphs by colorizing images, i.e., adding two more channels and adding independent Gaussian noise, drawn from $\mathcal{N}(0, 0.6)$, to each channel.

- **Graph-SST2** [10]: Each graph is converted from a text sequence, where nodes represent words, edges indicate relations between words, and label is the sentence sentiment. Graphs are split into different sets according to their average node degree to create distribution shifts. We use constant edge weights and filter out the graphs with edges less than three. The node features are initialized by the pre-trained BERT word embedding [11].

- **Open Graph Benchmark (OGB)** [4]: We consider two graph property prediction datasets with distribution shifts, i.e., OGBG-MOLSIDER and OGBG-MOLHIV. The task is to predict the target molecular properties. We adopt the default scaffold splitting procedure, i.e., splitting the graphs based on their two-dimensional structural frameworks. Note that this scaffold splitting strategy aims to separate structurally different molecules into different subsets, which provides a more realistic and challenging scenario for testing graph out-of-distribution generalization.

The real-world datasets are publicly available as follows:

- **MNIST-75sp**: `http://yann.lecun.com/exdb/mnist/` with license unspecified

- **Graph-SST2**: `https://github.com/divelab/DIG/tree/main/dig/xgraph/datasets` with GPL-3.0 License

- **Open Graph Benchmark (OGB)**: `https://ogb.stanford.edu/docs/graphprop/` with MIT License

## E.2 GNN Configurations

We summarize the backbone GNN for $GNN^{\mathbf{M}}$, $GNN^{\mathbf{V}}$, $GNN^{\mathbf{I}}$ and readout function $READOUT^{\mathbf{V}}$, $READOUT^{\mathbf{I}}$ in Table 3. These settings are set the same as [2] for a fair comparison. For $GNN^{\mathbf{M}}$, the number of layers is 2. $GNN^{\mathbf{V}}$ and $GNN^{\mathbf{I}}$ adopt shared parameters, and the number of layers is 4. The dimensionality of the graph-level and node-level representations $d$ is 300 for OGB, 128 for Graph-SST2, and 32 for other datasets.

Table 3: The backbone GNNs and global pooling function of each dataset.

| | SP-Motif | MNIST-75sp | Graph-SST2 | OGBG-MOLSIDER | OGBG-MOLHIV |
|---|---|---|---|---|---|
| $GNN^{\mathbf{M}}$/$GNN^{\mathbf{V}}$/$GNN^{\mathbf{I}}$ Backbone | Local Extremum GNN [12] | $k$-GNNs [13] | ARMA [14] | GIN + Virtual nodes [15, 4] | GIN + Virtual nodes [15, 4] |
| $READOUT^{\mathbf{I}}$/$READOUT^{\mathbf{V}}$ | Mean Pooling | Max Pooling | Mean Pooling | Add Pooling | Add Pooling |

## E.3 Baselines

We provide detailed descriptions and links to code repository of baselines in our experiments as follows:

- **ERM**: We use ERM to denote the backbone GNN models listed in Table 3, which are trained with the standard empirical risk minimizing.

- **Attention**[1] [16]: For this baseline, we replace the default layers of $\text{GNN}^{\mathbf{M}}$ into graph attention layers. $\text{GNN}^{\mathbf{I}}$ and $\text{GNN}^{\mathbf{V}}$ are kept the same as ERM, which adopt the default layers shown in Table 3.

- **Top-k Pool**[2] [17]: This method implements the regular global top-k pooling operation on graph data. It selects a subset of important nodes to enable high-level feature encoding and receptive field enlargement. We add this pooling layer after the last layer of $\text{GNN}^{\mathbf{M}}$.

- **SAG Pool**[3] [18]: It exploits the self-attention mechanism to distinguish between the nodes that should be neglected and the nodes that should be chosen genearting the subgraph. Thanks to the self-attention mechanism which uses graph convolutions to calculate attention scores, node features and graph topology are jointly considered. We add this pooling layer after the last layer of $\text{GNN}^{\mathbf{M}}$.

- **ASAP**[4] [12]: It adopts self-attention with a modified GNN formulation to identify the importance of nodes in the graph and learns a sparse soft cluster assignment for nodes at each layer to effectively pool the subgraphs.

- **GroupDRO**[5] [19]: It seeks to optimize the worst-performance over a distribution set to achieve OOD generalization performance.

- **IRM**[6] [20]: It is a representative invariant learning method, seeking to find data representations or features for which the optimal predictor is invariant across all environments. We conduct random environment partitions on the input graph datasets for training.

- **V-REx**[7] [21]: This method is proven to be able to recover the causal mechanisms of the targets and is robust to distribution shifts. Since this method relies on the explicit environment labels that are unavailable for the graph datasets in a mixture of latent environments, we conduct random environment partitions on the input graph datasets during the training stage.

- **DIR**[8] [2]: It conducts interventions on graphs to create interventional distributions and improve generalization.

- **GSAT**[9] [22]: It aims to build inherently interpretable GNNs and expects GNNs to be more generalizable by penalizing the amount of information from the input data.

### E.4  Additional Details of Optimization and Hyper-parameters

The number of epochs for optimizing our proposed method and baselines is set to 100. We adopt Stochastic Gradient Descent (SGD) for the optimization on Graph-SST2 and OGB datasets (the batch size is 32), and Gradient Descent (GD) for SP-Motif and MNIST-75sp, following the setting in [2] for a fair comparison. Each model is evaluated on the provided validation set for OGB or a holdout in-distribution validation set for the other datasets for each epoch. We adopt an early stopping strategy, i.e., stop training if the performance on the validation set does not improve for 5 epochs. Since we focus on graph classification tasks, we use the cross-entropy loss as the loss function $\ell$. The activation function is ReLU [23]. The evaluation metric is ROC-AUC for OGB datasets and accuracy for the others. In the invariant subgraph indentification module, the invariant subgraph generator selects $t \times |E|$ edges for each graph, i.e., $\text{Top}_t$ in Eq. (3), to generate the invariant subgraph. For a fair comparison, we uniformly set the hyper-parameter $t$ for our method and baselines as 0.25, 0.9, 0.6, and 0.8 on SP-Motif, MNIST-75sp, Graph-SST2, and two OGB datasets, respectively. The hyper-parameter $\lambda$ is chosen from $\{10^{-5}, 10^{-3}, 10^{-1}\}$ and the number of environments $|\mathcal{E}_{infer}|$ is chosen from $[2, 4]$ based on the results of the validation set. The selected $\lambda$ and $|\mathcal{E}_{infer}|$ are reported in Table 4.

---

[1] `https://github.com/PetarV-/GAT` with MIT License

[2] `https://github.com/HongyangGao/Graph-U-Nets` with GPL-3.0 License

[3] `https://github.com/inyeoplee77/SAGPool` with license unspecified

[4] `https://github.com/malllabiisc/ASAP` with Apache-2.0 License

[5] `https://github.com/kohpangwei/group_DRO` with license unspecified

[6] `https://github.com/facebookresearch/InvariantRiskMinimization` with Attribution-NonCommercial 4.0 International License

[7] `https://github.com/capybaralet/REx_code_release` with license unspecified

[8] `https://github.com/wuyxin/dir-gnn` with MIT License

[9] `https://github.com/Graph-COM/GSAT` with MIT License

Table 4: The chosen hyper-parameters of $\lambda$ and $|\mathcal{E}_{infer}|$ on each dataset.

| | SP-Motif | MNIST-75sp | Graph-SST2 | OGBG-MOLSIDER | OGBG-MOLHIV |
|---|---|---|---|---|---|
| $\lambda$ | $10^{-5}$ | $10^{-5}$ | $10^{-1}$ | $10^{-3}$ | $10^{-3}$ |
| $|\mathcal{E}_{infer}|$ | 3 | 2 | 2 | 2 | 2 |

## E.5 Additional Details on Silhouette Score

Silhouette score [24], a commonly used evaluation metric for clustering, is defined as the mean Silhouette coefficient over all samples. The Silhouette coefficient is calculated using the mean intra-cluster distance (denoted as $d_i$) and the mean nearest-cluster distance (denoted as $d_n$) for each sample. The Silhouette coefficient for a sample is $(d_n - d_i)/max(d_i, d_n)$. Therefore, Silhouette score falls within the range $[-1, 1]$. A silhouette score close to 1 means that the clusters become dense and nicely separated. The score close to 0 means that clusters are overlapping. And the score of smaller than 0 means that data belonging to clusters may be wrong/incorrect.

## E.6 Hardware and Software Configurations

We conduct the experiments with:

- Operating System: Ubuntu 18.04.1 LTS

- CPU: Intel(R) Xeon(R) CPU E5-2699 v4@2.20GHz

- GPU: NVIDIA GeForce GTX TITAN X with 12GB of Memory

- Software: Python 3.6.5; NumPy 1.19.2; PyTorch 1.10.1; PyTorch Geometric 2.0.3 [25].

## E.7 Additional Experiments on Environment Inference

### E.7.1 Experiments with Ground-truth Environments

We further conduct empirical analyses to investigate the performances of our model with the ground-truth environments. We compare our original model (**GIL**) with the model directly using the ground-truth environments (termed as **GIL** w. GT Env.) on the synthetic SP-Motif dataset ($r_{test} = 1/3$). The results in Table 5 show that the performance of using the inferred environments by our model and the ground-truth environments are comparable, even under different strengths of distribution shifts. The results are also expected since our inferred latent environments are largely aligned with the ground-truth labels, as shown in Figure 5 of the main paper. We think it would be interesting to conduct more explorations for real-world graphs when their environment labels are available.

Table 5: The accuracy (%) on SP-Motif ($r_{test} = 1/3$) when directly adopting the ground-truth environments (**GIL** w. GT Env.) compared with the original model (**GIL**).

| $r_{train}$ | $r = 1/3$ | $r = 0.5$ | $r = 0.6$ | $r = 0.7$ | $r = 0.8$ | $r = 0.9$ |
|---|---|---|---|---|---|---|
| **GIL** | $55.44_{\pm 3.11}$ | $54.56_{\pm 3.02}$ | $53.60_{\pm 4.82}$ | $53.12_{\pm 2.18}$ | $51.24_{\pm 3.88}$ | $46.04_{\pm 3.51}$ |
| **GIL** w. GT Env. | $55.42_{\pm 2.98}$ | $54.63_{\pm 3.10}$ | $53.58_{\pm 4.67}$ | $53.18_{\pm 3.02}$ | $51.01_{\pm 4.02}$ | $46.23_{\pm 3.16}$ |

### E.7.2 Experiments with Different Clustering Algorithms

As discussed in the main paper, we adopt the k-means clustering algorithm [26] to infer the environment labels. In addition, we explore another popular clustering algorithm [27] (termed as convex clustering) to infer the environment labels. The results on the synthetic SP-Motif dataset ($r_{test} = 1/3$) are shown in Table 6. These results indicate that the clustering algorithm could have a slight influence on the model performance and overall our model is not sensitive to the choice for clustering algorithm. Our proposed model does not rely on specific clustering algorithm to infer the environment labels and can also be compatible with other clustering algorithms.

Table 6: Environment inference with different clustering algorithms on the SP-Motif ($r_{test} = 1/3$).

| $r_{train}$ | $r = 1/3$ | $r = 0.5$ | $r = 0.6$ | $r = 0.7$ | $r = 0.8$ | $r = 0.9$ |
|---|---|---|---|---|---|---|
| **GIL** (k-means) | $55.44_{\pm3.11}$ | $54.56_{\pm3.02}$ | $53.60_{\pm4.82}$ | $53.12_{\pm2.18}$ | $51.24_{\pm3.88}$ | $46.04_{\pm3.51}$ |
| **GIL** (convex clustering) | $55.21_{\pm2.45}$ | $53.60_{\pm4.74}$ | $54.01_{\pm5.13}$ | $53.43_{\pm1.94}$ | $50.12_{\pm4.15}$ | $47.01_{\pm2.54}$ |

### E.8 Sensitivity of READOUT Functions and GNN Architectures

We conduct the sensitivity analysis on the READOUT functions and GNN architectures in Table 7. The results show that the choices of READOUT functions and GNN architectures have a slight influence on the performances. Overall, our model is not very sensitive to their choices and can be compatible with most common READOUT functions and GNN backbones.

Table 7: The performance (ROC-AUC, %) with different READOUT functions and GNN architectures.

|  | MOLSIDER | MOLHIV |
|---|---|---|
| GIN + add pooling | $63.50_{\pm0.57}$ | $79.08_{\pm0.54}$ |
| GIN + max pooling | $63.37_{\pm0.72}$ | $\mathbf{79.10_{\pm0.42}}$ |
| GIN + mean pooling | $61.91_{\pm0.75}$ | $78.16_{\pm0.47}$ |
| GCN + add pooling | $62.31_{\pm1.12}$ | $77.23_{\pm0.61}$ |
| GCN + max pooling | $\mathbf{63.68_{\pm0.91}}$ | $77.61_{\pm0.59}$ |
| GCN + mean pooling | $61.33_{\pm0.45}$ | $76.98_{\pm0.36}$ |