# OpenReview forum: "Learning Invariant Graph Representations for Out-of-Distribution Generalization"
_NeurIPS.cc/2022/Conference — NeurIPS 2022 Accept_

### Official Review · Reviewer_uGwK · 2022-07-09

**Rating:** 5
**Confidence:** 3
**Soundness:** 3 good
**Presentation:** 3 good
**Contribution:** 3 good

**Summary:**

This paper studies the out-of-distribution (OOD) generalization problem on graphs under a mixture of environments.

It proposes a graph invariant learning (GIL) solution to learn a maximally invariant graph predictor, which composes an environment inference module and an invariant subgraph identification module.

**Questions:**

As mentioned above, the main concern is about the novelty and lack of adequate literature discussions.

**Limitations:**

No obvious potential negative societal impact. This paper should also discuss a bit about the applicability of the assumptions made in this paper.

**Strengths And Weaknesses:**

Strengths:

1. The problem this paper studied is important, i.e.,  OOD generalization of GNNs, and it proposes an out-of-distribution generalization framework for GNNs, which composes environment inference and invariant subgraph identification/generalization.
2. The presentation logic flow is clear.

Weaknesses:

1. The novelty and significance of this paper are a concern as some important literature is not discussed in this paper. There are several works also studying OOD generalization on graphs but corresponding discussions are missing in the paper.
	1. Wu et al. 2022 [1] study the OOD generalization in node classification, and takes similar assumptions and solutions in this paper.
	2. Miao et al., 2022 [2] also discussed the application of graph information bottleneck criteria for OOD generalization.
2. Especially, in the literature on invariant learning, the key method in this paper shares many similarities with HRM[3], which should be discussed and compared substantially.
3. In experiments, [1,2] should all be included as baselines, and direct applying [3] to graph data should also be included as a baseline.


[1] Qitian Wu, Hengrui Zhang, Junchi Yan, David Wipf. Handling Distribution Shifts on Graphs: An Invariance Perspective. ICLR 2022.

[2] Siqi Miao, Miaoyuan Liu and Pan Li. Interpretable and Generalizable Graph Learning via Stochastic Attention Mechanism. ICML 2022.

[3] Jiashuo Liu, Zheyuan Hu, Peng Cui, Bo Li, and Zheyan Shen. Heterogeneous risk minimization. ICML 2021.

---

> ### Author Response · Authors · 2022-08-02
> **Response to Reviewer uGwK**
>
> We thank the reviewer for the valuable comments. Please kindly find the detailed responses below.
>
> +  **Q1. Some literature is not discussed in this paper.**
>
> **R1:**  Thank you for sharing with us these up-to-date works. We discuss the differences between these works and ours to clarify the novelty and significance of this paper as follows. EERM [1] focuses on **node-level** OOD prediction problem while our model studies graph-level OOD prediction problem, which are two different tasks (see Q3 for the experimental comparisons). GSAT [2] is a very recent **interpretable** graph learning method, which is just accepted by ICML2022. GSAT and our model have different basic assumptions and training schemes. Specifically, GSAT is mainly to build inherently interpretable GNNs and expect GNNs to be more generalizable by penalizing the amount of information from the input data. Our model, based on the invariance principle, targets directly learning invariant graph representations under a mixture of latent environments for OOD generalization.  We have cited these works and added relevant discussions in the revised version.
>
> We would also like to clarify the technical novelty and contributions of this work as follows. **(1)** We focus on a novel and challenging problem, i.e., learning invariant graph representations in a mixture of latent environments. To the best of our knowledge, this problem is not explored by the existing works and its formulation is also not well established in the literature due to the unique challenges for graphs under distribution shifts. **(2)** Since graph data usually comes from a mixture of latent environments, most existing invariance regularizers rely on accurate environment labels, so they cannot be directly applied to graphs. It also remains a new research direction to explore the environments on graphs based on the environment-discriminative (variant) graph patterns and encourage the graph encoder to capture environment-agnostic (invariant)  graph patterns more accurately for OOD generalization. **(3)** The environment inference module and invariant learning module can mutually promote each other by focusing on two complementary parts of the input graphs, i.e., variant subgraphs and invariant subgraphs, respectively. Identifying the invariant graph patterns among latent environments raise unique and critical challenges. **(4)**  We design a theoretically grounded learning scheme to find a maximal invariant subgraph generator for solving the graph OOD generalization problem. Our model achieves performance gains on both synthetic and real-world benchmarks with significant improvements against various baselines. We believe these technical contributions are non-trivial and have potential impacts to the graph community.
>
> +  **Q2. Missing the invariant learning literature [3].**
>
>  **R2:** Thank you for this suggestion. The key difference between [3] and ours is that although the work [3] considers inferring latent environments during learning process, it is designed for dealing with the simple scenario where data is raw feature as that paper claims, while the problems of jointly conducting environment inference and invariant learning on graph-structured data remains unexplored. We further formulate the problem as finding maximal invariant subgraph in a mixture of latent environments and propose a corresponding algorithm for graphs whose effectiveness is demonstrated in theory and practice. Detailed experimental comparisons are provided in the response to the following question.

---

> > ### Author Response · Authors · 2022-08-02
> > **Response to Reviewer uGwK (Cont.)**
> >
> > +  **Q3. Adding baselines including EERM, GSAT, and HRM to the experiments.**
> >
> > **R3:**  Thank you for the comment. We have added comparisons with the methods mentioned above. Notice that EERM is not designed for graph-level OOD generalization problems,  so we have modified its graph structure editers to create graph-level multiple environments.
> >
> > The results on the synthetic dataset SP-Motif (Scenario 1: $r_{test}=1/3$) are as follows.
> >
> > | $r_{train}$                 |   $r=1/3$    |   $r=0.5$    |   $r=0.6$    |   $r=0.7$    | $r=0.8$    |   $r=0.9$    |
> > | ---- | :--: | :--: | :--: | :--: | :--: | :--: |
> > | HRM  | 51.43 ± 4.08  | 50.34 ± 3.96 | 46.34 ± 5.91 | 38.94 ± 4.12  | 38.20 ± 3.71 | 37.10 ± 4.80 |
> > | EERM | 52.89 ± 3.20 | 51.97 ± 2.87 | 50.87 ± 4.97 | 45.38 ± 2.90  | 42.98 ± 3.63 | 42.42 ± 3.67 |
> > | GSAT | 53.67 ± 3.65  | 53.34 ± 4.08 | 51.54 ± 3.78 | 50.12 ± 3.29 | 45.83 ± 4.01 | 44.22 ± 5.57 |
> > | GIL  | 55.44 ± 3.11  | 54.56 ± 3.02 | 53.60 ± 4.82 | 53.12 ± 2.18  | 51.24 ± 3.88 | 46.04 ± 3.51 |
> >
> > The results on the synthetic dataset SP-Motif (Scenario 2: $r_{test}=0.2$) are as follows.
> >
> > | $r_{train}$                 |   $r=1/3$    |   $r=0.5$    |   $r=0.6$    |   $r=0.7$    | $r=0.8$    |   $r=0.9$    |
> > | ---- | :--: | :--: | :--: | :--: | :--: | :--: |
> > | HRM  | 51.79 ± 3.18 | 40.91 ± 4.06 | 35.89 ± 5.10 | 33.08 ± 4.11 | 28.18 ± 2.93 | 23.65 ± 4.12 |
> > | EERM | 51.07 ± 2.75 | 48.65 ± 3.65 | 44.28 ± 5.02 | 38.39 ± 4.91 | 33.01 ± 3.89 | 31.08 ± 2.85 |
> > | GSAT | 51.36 ± 4.21 | 50.48 ± 3.98 | 46.93 ± 5.03 | 43.55 ± 3.67 | 40.35 ± 4.21 | 33.87 ± 5.19 |
> > | GIL  | 54.80 ± 3.93   | 52.48 ± 4.41   | 50.08 ± 5.47   | 47.44 ± 2.87   | 46.36 ± 3.80   | 35.80 ± 5.03   |
> >
> >
> > The results on the real-world datasets are as follows.
> >
> > |      |  MNIST-75sp  |  Graph-SST2  |   MOLSIDER   |    MOLHIV    |
> > | ---- | :----------: | :----------: | :----------: | :----------: |
> > | EERM | 17.13 ± 1.89 | 81.39 ± 0.63 | 58.92 ± 1.03 | 76.27 ± 1.48 |
> > | HRM  | 17.05 ± 2.84 | 81.15 ± 0.71 | 57.41 ± 1.86 | 75.12 ± 1.17 |
> > | GSAT | 20.12 ± 1.35 | 82.95 ± 0.58 | 60.82 ± 1.36 | 76.47 ± 1.53 |
> > | GIL  | 21.94 ± 0.38 | 83.44 ± 0.37 | 63.50 ± 0.57 | 79.08 ± 0.54 |
> >
> > Besides the results above, we **have also updated the results** of comparisons shown in Figures 3, 6 of the main paper and Figures 4, 5, 6 of Appendix in the revised paper.
> >
> > These results show that: (1) Although outperforming ERM on most datasets, EERM [1], as one node-level OOD generalization method, cannot well handle graph-level distribution shifts for promising results. (2) Directly adopting HRM [3] that is proposed for raw feature data on more complex graph-structured data produces poor OOD generalization performance. (3) GSAT [2], one very recent baseline, achieves competitive results in most comparisons. Nevertheless, **our proposed method still consistently achieves the best performance**, demonstrating the effectiveness of the proposed method for learning invariant graph representations under a mixture of latent environments. We have added these comparisons to the revised paper.
> >
> > +  **Q4. The applicability of the assumptions.**
> >
> > **R4:** We believe the assumptions can be easily satisfied for various types of real-world graphs. For Assumption 3.1, it was reasonably introduced in [4] and can be verified for real-world graphs. For example, as present in Figure 1 in Appendix, for molecule graphs labeled by specific properties [5, 6], the optimal invariant subgraphs represent the "hydrophilic R-OH group"/"non-polar repeated ring structures", whose relationship with the label solubility/anti-solubility is truly predictive and invariant across different environments. And the corresponding variant subgraphs denote the shared carbon structure  or scaffold [5, 6], which could change across different environments.  Similarly, for superpixel graphs from [4], invariant and variant subgraphs represent the edges corresponding to the digit itself and other edges from the background, respectively. The label is only dependent on the digit and the background information could change while having no influence on the labels. And for the Graph-SST2 datasets (as shown in Figure 7 in the main paper),  invariant subgraphs are the positive/negative words in the sentences that are salient for sentiment labels. Therefore, the assumptions commonly hold in real world.
> >
> >
> >
> > **References:**
> >
> > [1] Wu et al., Handling Distribution Shifts on Graphs: An Invariance Perspective. ICLR, 2022.
> >
> > [2] Miao et al., Interpretable and Generalizable Graph Learning via Stochastic Attention Mechanism. ICML, 2022.
> >
> > [3] Liu et al., Heterogeneous risk minimization. ICML, 2021.
> >
> > [4] Wu et al., Discovering Invariant Rationales for Graph Neural Networks. ICLR, 2022.
> >
> > [5] Hu et al., Open Graph Benchmark. NeurIPS, 2020.
> >
> > [6] Duvenaud et al., Convolutional networks on graphs for learning molecular fingerprints. NeurIPS, 2015.

---

> ### Author Response · Authors · 2022-08-08
> **Kind reminder to the reviewer**
>
> Dear Reviewer,
>
> We have provided detailed responses to your comments. We are wondering whether your concerns have been properly addressed.
>
> If you have further questions after reading the responses and the revised paper, it would be great to let us know. We are happy to address them.
>
>
> Best regards,
>
> The Authors

---

> ### Comment · Reviewer_uGwK · 2022-08-09
> **Thanks for detailed responses**
>
> Thanks for the authors' detailed responses. My concerns about the literature and experiments are resolved. I would like to raise my score to 5.

---

> > ### Author Response · Authors · 2022-08-09
> > **Thank you for the quick reply**
> >
> > Dear Reviewer,
> >
> > We sincerely appreciate your quick response! And thanks again for your time and efforts in reviewing our work.
> >
> > Best regards,
> >
> > The Authors

---

### Official Review · Reviewer_52ur · 2022-07-11

**Rating:** 6
**Confidence:** 4
**Soundness:** 4 excellent
**Presentation:** 3 good
**Contribution:** 2 fair

**Summary:**

In this work the authors propose a three part method for learning graph representations that are robust to distribution shifts. First their method identifies _invariant_ subgraphs, then clusters the _variant_ complements to the invariant subgraphs to infer latent environment codes, and finally learns a representation function and linear predictor over the extracted invariant subgraphs and latent environment codes. This system is optimized end to end and they perform an empirical evaluation over synthetic and real world data to demonstrate the effectiveness of the approach.

**Questions:**

1. Relating to the comment above about environments sharing subgraph structure, this is very much a broad assumption in the entire work. It is _fine_ because though it is a narrow focus it is a fundamental type of distribution shift for graphs. However, it feels like conditions such as $P^{e}(\phi(G)) = P^{e'}(\phi(G))$ would generally be too strong in practical settings. Especially for the real world data, did you attempt to empirically verify that some of the conditions for Thm 4.1 approximately hold?

2. Can you elaborate on why the number of clusters for the k-means component was limited to $[2,4]$? Having to explicitly choose this parameter is one of the obvious points of difficulty for the model with datasets comprising an unknown number of latent environments.

**Limitations:**

The few limitations are relatively minor:

1. Methodological focus on shared subgraph/scaffold structure between environments
2. System bottleneck on the effectiveness of the unsupervised environment identification module. Ideally they would explore different choices for the clustering algorithm.
3. The analysis in Figures 4 and 5 of Silhouette score and the TSNE projection of clusters should be extended beyond the synthetic dataset where the environments are perfectly separable, as the claims they were used to support may not be very generalizable.
4. Hyperparameter sensitivity analysis should be extended beyond the synthetic dataset, even though it is plausible that the trends generalize.

**Strengths And Weaknesses:**

### Strengths

**Quality**: The empirical evaluation is well scoped in its choice of datasets as well as the diversity of the shift parameter settings and types of analysis showcased. In particular the SP-Motif and OGB selections are well suited to their method's focus on shared subgraph structures. The agreement between the performance rankings of GIL/DIR on SP-Motif and MOLHIV is a good sign since shared subgraphs and scaffolds are similar concepts. The finding that ERM generally outperforms most methods matches prior work for MOLHIV and this then underscores the performance of GIL (and the competitor DIR). Finally, the fact that the performance of the method is also demonstrated on datasets derived from different underlying signals (MNIST and SST2) that are not molecule based, serves as a good generality sanity check.

**Clarity**: The combination of Figure 2 and Section 3 describing their method and training procedures is well presented and easy to understand. Section 5 is broken down well into its subsections and the figures 6 and 7 are appreciated as visual references of the in/variant structures being identified.

**Significance**: Performance suggests that the approach is minimally competitive, and potentially SOTA pushing on the OGB data, though the way in which SP-Motif is used at the sweep of parameter settings makes it slightly harder to directly compare to other work. Method is simpler than some competitor approaches for OOD generalization, which is preferred.

### Weaknesses

**Clarity**: Theorems in Section 4, especially Thm 4.2 can likely be moved to the Appendix, as they don't add very much to the impact of the work compared to the empirical results. Further, Thm 4.1, especially the RHS, is stated very generally rather than in the specific setting of subgraph-based inter-environment invariance, which is not the _only_ kind of environment differentiation one expects in real world data (i.e. feature shift or label imbalance etc. are also options - these are in fact explored in the empirical results).

Work would generally benefit from a close editing pass from a native english speaker, _but this does not factor into my assessment_.

**Originality/Significance**: The three components of the method are not particularly novel on their own - in particular, the invariant regularized learner is very derivative of IRM in its objective. As such the work overall feels like a DL engineering solution for an E2E system, which is not without value by any means, but overall not very methodologically novel.

---

> ### Author Response · Authors · 2022-08-02
> **Response to Reviewer 52ur (Part 1/3)**
>
> We thank the reviewer for the valuable positive feedback. We addressed all the comments. Please kindly find the detailed responses below.
>
> +  **Q1.1.  Thm 4.2 can likely be moved to the Appendix.**
>
> **R1.1:** Thank you for this suggestion. We have moved Thm 4.2 into Appendix in the revised version.
>
> +  **Q1.2.  Clarifications on Thm 4.1. It is stated generally rather than in the specific setting.**
>
> **R1.2:** We would like to explain Thm 4.1 for better clarification. The RHS of Eq. (10) in Thm 4.1 is the objective for OOD generalization via the invariance principle [1-4], which is also described in the problem formulation (Eq. (1) in Section 2). It means that we aim to solve the OOD problem by learning invariant predictors that can generalize across environments. In this paper, we study invariant learning for graph representation learning under a mixture of latent environments. Considering the challenges on graphs (as present in lines 45-53 in Section introduction), **we transform the graph OOD generalization problem into finding the optimal invariant subgraphs**, which is indicated by the LHS of Eq. (10) in Thm 4.1, showing that our proposed method effectively solves our targeted problem, i.e., LHS = RHS.
>
> In addition, since the distribution shifts on graphs could exist in both feature-level and structure-level, we expect that our model through finding the optimal invariant subgraphs can handle general and diverse distribution shifts. The empirical experiments validate that our model indeed can achieve good performance on various types of distribution shifts.
>
> + **Q2. The three components of the method are not particularly novel on their own.**
>
> **R2:** Thank you for this comment. We would like to clarify the  technical novelty and contributions of this work as follows. **(1)** We focus on a novel and challenging problem, i.e., learning invariant graph representations in a mixture of latent environments. To the best of our knowledge, this problem is not explored by the existing works and its formulation is also not well established in the literature due to the unique challenges for graphs under distribution shifts. **(2)** Since graph data usually comes from a mixture of latent environments and most existing invariance regularizers rely on accurate environment labels, they cannot be directly applied to graphs. It also remains a new research direction and is more challenging to identify the complex invariant patterns on graphs among latent environments. **(3)** Our model non-trivially fuses the advantages of the three components rather than simply combining them through engineering. The environment inference module and invariant learning module focus on two complementary parts of the input graphs, i,e., variant subgraphs and invariant subgraphs, respectively. Therefore, the two modules can mutually promote each other to identify more accurate invariant and variant patterns during training process. This is also a new insight for graph OOD generalization. **(4)**  We design a theoretically grounded learning scheme to find the maximal invariant subgraph generator for solving the graph OOD generalization problem. Our model achieves performance gains on both synthetic and real-world benchmarks with significant improvements against various baselines. We believe these technical contributions are non-trivial and have potential impacts to the community.

---

> > ### Author Response · Authors · 2022-08-02
> > **Response to Reviewer 52ur (Part 2/3)**
> >
> > +  **Q3. Can the conditions for Thm 4.1 approximately hold?**
> >
> > **R3:** Thank you for this question. We would like to clarify that the conditions of Thm 4.1 can approximately hold in real world.
> >
> > The first condition of Thm 4.1 means that **the optimal invariant subgraph and optimal variant subgraph  (i.e., the complement of the optimal invariant subgraph) should be independent** in ideal situations (although we may observe correlations between them due to the bias in datasets). This condition is  widely assumed in the invariant learning literature  [1-5] and also commonly satisfied for real-world graphs. For example, as present in Figure 1 in Appendix, for molecule graphs labeled by specific properties [6, 7], the optimal invariant subgraphs represent the "hydrophilic R-OH group"/"non-polar repeated ring structures", whose relationship with the label solubility/anti-solubility is truly predictive and invariant across different environments. And the corresponding variant subgraphs denote the shared carbon structure or scaffold [6, 7], which are independent of the subgraphs reflecting the properties.  For superpixel graphs from [5], invariant and variant subgraphs represent the edges corresponding to the digit itself and other edges from the background, respectively, where the digit and background information are also independent. Therefore, the first condition of Thm 4.1 commonly holds in real world.
> >
> > The second condition of Thm 4.1 means that **the distribution of  training graphs consists of  enough diverse environments**. This condition is also widely adopted in the literature [1-5] for practical solutions. For example, as one of the representative baselines in our comparisons, DIR [5] conducts interventions on the training distribution to create multiple interventional distributions. Likewise, for the real-world molecule graphs,  there exist some samples whose invariant subgraphs (i.e., the part reflecting specific properties) are the same but variant subgraphs (i.e., carbon structure or scaffold) are different, rather than one type of invariant subgraph only has unique and single type of variant subgraph.
> >
> > + **Q4. Why the number of clusters for the k-means is chosen from [2, 4]?**
> >
> > **R4:** Thank you for this question. The number of clusters $|\mathcal{E}\_{infer}|$ in Eq. (5) is a hyper-parameter, whose sensitivity analysis **in a large range** is provided in Section 5.5. Overall, although it is an important hyper-parameter to the performance, our model is not very sensitive to the choice of cluster number. From Figure 8(a), we can observe that our model does not need to be specified the ground truth number of latent environments and can achieve promising results in a wide range of this hyperparameter. Although the performance can reach a peak when $|\mathcal{E}\_{infer}|$ matches the ground truth, our model still outperforms the most competitive baselines when $|\mathcal{E}\_{infer}|$ does not equal the ground truth or when the ground truth is unknown. We leave the automatic search for the best choice of this hyper-parameter as the future work.
> >
> > +  **Q5. Methodological focus on shared subgraph/scaffold structure between environments.**
> >
> > **R5:** Thank you for this comment. In this work, we capture environment-agnostic (i.e., invariant among environments) graph patterns to achieve good OOD generalization. We find that this assumption is widely adopted in the literature [1-5] and also commonly satisfied for real-world graphs. For example, as present in Figure 1 in Appendix, for molecule graphs labeled by specific properties, the invariant subgraphs represent the "hydrophilic R-OH group"/"non-polar repeated ring structures", whose relationship with the label solubility/anti-solubility is truly predictive and invariant across different environments. And variant subgraphs denote the shared carbon structure or scaffold [6, 7]. For superpixel graphs from [5], invariant and variant subgraphs represent the edges corresponding to the digit itself and other edges from the background, respectively.

---

> > > ### Author Response · Authors · 2022-08-02
> > > **Response to Reviewer 52ur (Part 3/3)**
> > >
> > > +  **Q6. Explore different choices for the clustering algorithm.**
> > >
> > > **R6:** In this work, we use the k-means clustering algorithm to infer the environment labels. Following your suggestion, we compare with another popular clustering algorithm (termed as convex clustering) proposed in (Lashkari et al., Convex Clustering with Exemplar-Based Models. NeurIPS, 2007) to infer the environment labels.
> > >
> > > The results on the synthetic SP-Motif dataset ($r_{test}=1/3$) are as follows:
> > >
> > > | $r_{train}$                 |   $r=1/3$    |   $r=0.5$    |   $r=0.6$    |   $r=0.7$    | $r=0.8$    |   $r=0.9$    |
> > > | --------------------------- | :----------: | :----------: | :----------: | :----------: |:----------: | :----------: |
> > > | GIL (k-means)            | 55.44 ± 3.11 | 54.56 ± 3.02 | 53.60 ± 4.82 |53.12 ± 2.18| 51.24 ± 3.88 | 46.04 ± 3.51 |
> > > | GIL (convex clustering) | 55.21 ± 2.45 | 53.60 ± 4.74| 54.01 ± 5.13 | 53.43 ± 1.94 | 50.12 ± 4.15 |47.01 ± 2.54 |
> > >
> > >
> > > The results on the real-world datasets  are as follows:
> > >
> > >
> > > |                             |  MNIST-75sp  |  Graph-SST2  |   MOLSIDER   |    MOLHIV    |
> > > | --------------------------- | :----------: | :----------: | :----------: | :----------: |
> > > | GIL (k-means)            | 21.94 ± 0.38 | 83.44 ± 0.37 | 63.50 ± 0.57 | 79.08 ± 0.54 |
> > > | GIL (convex clustering) | 19.98 ± 0.57 | 82.89 ± 0.53 | 63.67 ± 0.43 | 79.01 ± 0.61 |
> > >
> > >
> > > These results show that the clustering algorithm could have a slight influence on the model performance and overall our model is not sensitive to the choice for clustering algorithm. It means that our model does not rely on specific clustering algorithm to infer the environment labels and can also be compatible with other clustering algorithms. We have added the results and analyses in Appendix E.7.3 of the revised paper.
> > >
> > > + **Q7. Clustering results beyond the synthetic dataset.**
> > >
> > > **R7:** Thank you for this suggestion. Following your suggestion, we add the Silhouette score during the training process on the MNIST-75sp dataset in Figure 2 in Appendix of the revised paper. We observe the similar pattern on MNIST-75sp dataset with the results on SP-Motif shown in Figure 4 in the main paper. The test accuracy and the clustering performance improve synchronously over training, indicating that the environment inference module and invariant learning module of our model can mutually enhance each other in both synthetic and real-world scenarios, which is one of the technical contributions of this paper.
> > >
> > > -  **Q8. Hyperparameter sensitivity analysis beyond the synthetic dataset.**
> > >
> > > **R8:**  Thank you for this comment. In Section 5.5, the hyperparameter sensitivity analyses are conducted on the synthetic dataset and one real-world dataset MNIST-75sp. Following your suggestion, we add more analyses on the real-world dataset MOLSIDER to study the sensitivity of hyper-parameters: the number of inferred environments $|\mathcal{E}\_{infer}|$, the regularizer coefficient $\lambda$, and the invariant subgraph mask size $t$.
> > >
> > > | $\|\mathcal{E}\_{infer}\|$ |       2       |      3       |      4       |      5       |      6       |
> > > | ----------------------- | :-----------: | :----------: | :----------: | :----------: | :----------: |
> > > | GIL                 | 63.50 ±  0.57 | 63.65 ± 0.39 | 63.88 ± 0.51 | 63.71 ± 0.41 | 63.72 ± 0.33 |
> > >
> > > | $\lambda$ | **$10^{-6}$** | $10^{-5}$ | $10^{-4}$ | $10^{-3}$ | $10^{-2}$ | $10^{-1}$ | $10^{0}$ |
> > > | --------- | :-------: | :-------: | :-------: | :-------: | :-------: | :-------: | :-------: |
> > > | GIL   | 61.71 ± 0.29 | 62.24 ± 0.32 | 63.76 ± 0.38 | 63.50 ± 0.57 | 62.87 ± 0.43 | 62.30 ± 0.39 | 62.11 ± 0.53 |
> > >
> > > | $t$     |     0.75     |     0.80     |     0.85     |     0.90     |     0.95     |
> > > | ------- | :----------: | :----------: | :----------: | :----------: | :----------: |
> > > | GIL | 63.10 ± 0.63 | 63.50 ± 0.57 | 63.46 ± 0.41 | 62.81 ± 0.33 | 62.13 ± 0.54 |
> > >
> > > We observe **similar patterns on this dataset with the results in Figure 8** of the main paper. **(1)** The number of environments has a slight impact on the model performance, indicating that our method is not sensitive to the number of inferred environments. **(2)** The coefficient $\lambda$  also has a slight influence by balancing the classification loss and the invariance regularizer term. **(3)** A very large mask size $t$ will result in too many edges in the invariant subgraph and bring in variant structures, while a small $t$ may let the invariant subgraph become too small to capture enough structural information. Overall, our model can outperform the best baselines with a wide range of hyper-parameters choices.

---

> > > > ### Author Response · Authors · 2022-08-02
> > > > **References**
> > > >
> > > > **References:**
> > > >
> > > > [1] Arjovsky et al., Invariant risk minimization. Arxiv, 2019.
> > > >
> > > > [2] Koyama et al., When is invariance useful in an Out-of-Distribution Generalization problem? Arxiv, 2020.
> > > >
> > > > [3] Krueger et al., Out-of-Distribution Generalization via Risk Extrapolation (REx). ICML, 2021.
> > > >
> > > > [4] Chang et al.,  Invariant Rationalization. ICML, 2020.
> > > >
> > > > [5] Wu et al., Discovering Invariant Rationales for Graph Neural Networks. ICLR, 2022.
> > > >
> > > > [6] Hu et al., Open Graph Benchmark. NeurIPS, 2020.
> > > >
> > > > [7] Duvenaud et al., Convolutional networks on graphs for learning molecular fingerprints. NeurIPS, 2015.

---

> ### Author Response · Authors · 2022-08-08
> **Kind reminder to the reviewer**
>
> Dear Reviewer,
>
> We have provided detailed responses to your comments. We are wondering whether your concerns have been properly addressed.
>
> If you have further questions after reading the responses and the revised paper, it would be great to let us know. We are happy to address them.
>
>
> Best regards,
>
> The Authors

---

> ### Comment · Reviewer_52ur · 2022-08-09
> **Acknowledgement of rebuttal by authors**
>
> Thank you for the detailed responses to all points made in the review.
>
> We have differing opinions on the validity of certain conditions in the OOD generalization and invariant learning problem literature, but your claim that these are not your own new assumptions, and rather shared with previous work is valid. The methodological novelty is of course also a slightly subjective matter but I will suggest that the specific language and four point enumeration used to state your rebuttal for Q2 below, is in my opinion slightly stronger than the original language, say, at the end of the introduction! Maybe you can incorporate parts of it into the revision. The additional experiments based on exploring the insensitivity to clustering algorithm and hyperparameters are also appreciated.

---

> > ### Author Response · Authors · 2022-08-09
> > **Thank you for your acknowledgement of our rebuttal**
> >
> > Dear Reviewer,
> >
> > We sincerely appreciate your positive feedback and acknowledgement of our rebuttal. And thanks again for your time and efforts in reviewing our work.
> >
> > Best regards,
> >
> > The Authors

---

### Official Review · Reviewer_PXQD · 2022-07-16

**Rating:** 5
**Confidence:** 3
**Soundness:** 2 fair
**Presentation:** 2 fair
**Contribution:** 2 fair

**Summary:**

This paper studies OOD generalization with a mixture of graph environments without environment labels. The method first generates the invariant subgraph and the variant subgraph based using a graph generator model. It then infers the environment labels by conducting k-means clustering on the variant subgraphs. This step assumes that the environment labels are correlated with variant subgraphs and are irrelevant to the invariant subgraph. The method then conducts invariant learning optimization across the different inferred environments.

**Questions:**

As shown in [1], the ground truth environment partition does not always lead to the best OOD generalization performance. Is this also true in the graph domain? If yes, how does the proposed kmeans clustering lead to better environment partitions for OOD generalization?

Reference:
[1] Environment Inference for Invariant Learning. In ICML 2021.

**Limitations:**

The authors have well addressed the limitations in the paper.

**Strengths And Weaknesses:**

This paper firstly studies OOD generalization for graphs without environment labels. The problem is clearly defined and discussed in the paper. Based on the assumptions, the author proposes a method that aligns with the identified challenges. The method later shows significant improvement over the baselines.

As for the weakness, I find gaps between the steps in theoretical analysis. In Equation (7), the optimization uses the ground truth environment $\mathcal{E}$, while Equation (8) uses the inferred environments $\mathcal{E}\_{infer} $ . The difference between $\mathcal{E}$ and $\mathcal{E}\_{infer} $ should be discussed to understand the proposed algorithm. The discussion should answer questions like "does $\mathcal{E}\_{infer}$ approximate $\mathcal{E}$?" and "what are the properties of $\mathcal{E}_{infer}$ for good OOD generalization performance?"

---

> ### Author Response · Authors · 2022-08-02
> **Response to Reviewer PXQD**
>
> We thank the reviewer for the insightful comments. Please kindly find the detailed responses to the comments below.
>
> +  **Q1.1. The difference between $\mathcal{E}$ and $\mathcal{E}\_{infer}$.**
>
> **R1.1:**  Thank you for this comment. We would like to clarify the difference and connection between $\mathcal{E}$ and $\mathcal{E}\_{infer}$. According to the definitions,  $\mathcal{E}$ is a random variable on indices of the ground-truth environments that are latent, and $\mathcal{E}\_{infer}$ is a random variable on indices of the inferred environments. We follow the invariant learning literature [1] to define invariant subgraph generator set $\mathcal{I}$ with respect to the ground-truth environment $\mathcal{E}$ and further derive the Theorem 3.2 (i.e., Equation (7)). However, it is often impossible to characterize such latent ground-truth environments in practice, which is the common issue in the invariant learning literature [1-4]. One common practical solution is to infer the latent environments from data and further assume that the model capable of  generalization across the inferred environments $\mathcal{E}\_{infer}$ can also generalize to the ground-truth environments. For example, the work [3] studies to discover environment labels by maximally violating the invariance principle and the work [4] proposes to create interventional distributions for generating multiple environments. Following similar schemes, we infer the latent environments $\mathcal{E}_{infer}$ with the representations of variant subgraph and propose the invariant learning module across the inferred environments for invariant and accurate predictions.
>
> +  **Q1.2. Does $\mathcal{E}\_{infer}$ approximate $\mathcal{E}$? When does $\mathcal{E}\_{infer}$ achieve good OOD generalization?**
>
> **R1.2:** Since we do not have ground-truth for $\mathcal{E}$, it is infeasible to directly constrain $\mathcal{E}\_{infer}$ to approximate $\mathcal{E}$. Instead, we require $\mathcal{E}\_{infer}$ to be inferred by only capturing the ground-truth environment-discriminative features while leaving the environment-agnostic (invariant) features. The model can achieve good OOD generalization performance under this requirement, which is consistent with the invariant learning literature [1-4]. We also observe from Figure 4 in the experiments that when invariant subgraphs are accurately discovered, the inference of latent environments can also be promoted by better capturing the environment-discriminate features which further enhances learning invariant subgraphs.  The two modules mutually enhance each other, leading to good OOD generalization performance in the experiments. These empirical results show the reasonableness and feasibility of this scheme.
>
> +  **Q2. Whether the given environment partition always leads to the best OOD generalization performance.**
>
> **R2:** Thanks for this insightful comment. We conduct empirical analyses to investigate this problem. Since the ground-truth environment labels are unavailable for the real-world datasets, we compare our original model (GIL with $\mathcal{E}\_{infer}$) with the model directly using the ground-truth environments (GIL with $\mathcal{E}$) on the synthetic SP-Motif dataset ($r_{test}=1/3$). The results are as follows.
>
> | $r_{train}$                     |   $r=1/3$    |   $r=0.5$    |   $r=0.6$    |   $r=0.7$    |   $r=0.8$    |   $r=0.9$    |
> | ------------------------------- | :----------: | :----------: | :----------: | :----------: | :----------: | :----------: |
> | GIL with $\mathcal{E}\_{infer}$ | 55.44 ± 3.11 | 54.56 ± 3.02 | 53.60 ± 4.82 | 53.12 ± 2.18 | 51.24 ± 3.88 | 46.04 ± 3.51 |
> | GIL with $\mathcal{E}$          | 55.42 ± 2.98 | 54.63 ± 3.10 | 53.58 ± 4.67 | 53.18 ± 3.02 | 51.01± 4.02  | 46.23 ± 3.16 |
>
> We can observe that **the performance of using the inferred environments by our model and the ground-truth environments are comparable**, even under different strengths of distribution shifts. The results are also expected since our inferred latent environments are largely aligned with the ground-truth labels, as shown in Figure 5 of the main paper.  Nevertheless, we think it would be interesting to explore this problem further for real-world graphs when their environment labels are available. We have added the analyses above into the Appendix  E.7.2 of the revised paper.
>
>
>
> **References:**
>
> [1] Koyama et al., When is invariance useful in an Out-of-Distribution Generalization problem? Arxiv, 2020.
>
> [2] Arjovsky et al., Invariant risk minimization. Arxiv, 2019.
>
> [3] Creager et al., Environment Inference for Invariant Learning. ICML, 2021.
>
> [4] Wu et al., Discovering Invariant Rationales for Graph Neural Networks. ICLR, 2022.

---

> ### Author Response · Authors · 2022-08-08
> **Kind reminder to the reviewer**
>
> Dear Reviewer,
>
> We have provided detailed responses to your comments. We are wondering whether your concerns have been properly addressed.
>
> If you have further questions after reading the responses and the revised paper, it would be great to let us know. We are happy to address them.
>
>
> Best regards,
>
> The Authors

---

### Official Review · Reviewer_oeB7 · 2022-07-17

**Rating:** 5
**Confidence:** 3
**Soundness:** 2 fair
**Presentation:** 3 good
**Contribution:** 2 fair

**Summary:**

The paper investigates a new research problem of learning invariant graph representations under distribution shifts by considering the latent environment labels. The proposed method, graph invariant learning (GIL), is a joint learning framework combing three different GNNs of various functions. With good empirical results on several datasets and related theoretical analyses, the paper justifies the effectiveness of the proposed GIL.

**Questions:**

1. It might be somehow confusing for the invariance property in Assumption 3.1 for its connection with the true labels of graphs. Two graphs of different environments are likely to be classified correctly even though they are not perfectly samples. So, what is the main difference between the invariance property and  $P^e(\Phi^*(G))=P^{e^{\prime}}(\Phi^*(G))$ as in the last condition of Theorem 4.1? Is it harder to be achieved? Or, can it provide a better guarantee?
2. The optimization process of the three used GNNs is unclear, which could be an essential design of the proposed system. I.e., how do the GNN$^M$, GNN$^V$, and GNN$^I$ be jointly optimized?
3. Besides, since the adjacent matrixes $A_I$ and $A_V$ are discrete, how are the gradients calculated and back propagated among these GNNs in an end-to-end manner?
4. The clustering effectiveness in Figure 5 seems quite perfect for its accurate partition, but the Score in Figure 4 is lower than 0.75. So, where is the gap come from? It would be better for the paper to introduce more about the Silhouette score, e.g., its basic calculation and range of values.
5. Regarding the prediction of latent environment labels are one of the main contributions, I would suggest the paper show the clustering cases without environment inference to better justify the effectiveness of such a designed module.

**Limitations:**

The READOUT functions, which shall be important for the proposed method, are not fully discussed and compared in the paper, including in the appendix part. Since the paper has mentioned the usage and the desired properties of READOUT functions many times, it would be better for the paper to provide a further discussion and related ablations. Besides, the influence of different GNN architectures, which served as backbones in the paper, is also missing.

**Strengths And Weaknesses:**

Pros

- The investigated problem is new and challenging, especially for real-world graph learning scenarios. With the well-defined research problem, the paper proposed to take the latent environment labels into consideration, which sheds an interesting direction on the current graph learning area.
- The gained empirical improvement is quite significant with the proposed GIL framework. Several ablation studies are shown.
- The paper is with good mathematical groundings for its provided theoretical analysis. Complexity analysis is also presented, including every single module of the proposed GIL framework.

Cons

- The importance of the latent environment labels is not fully justified, which should be the major contribution of the paper. It seems unclear for its motivation and significance and what benefits it can bring. In addition to the hyper-parameter study, there lacks instruction to extract, utilize, and analyze the latent environment labels.
- The technical novelty and contributions of the paper are neutral. The designed yet complicated framework seems to be the combination of several existing methods like the k-means and the invariance regularizer without presenting more insights into the new problem.

---

> ### Author Response · Authors · 2022-08-02
> **Response to Reviewer oeB7 (Part 1/4)**
>
> We thank the reviewer for the valuable feedback. We addressed all the comments. Please kindly find the detailed responses to the comments below.
>
> +  **Q1.1. The motivation and significance of inferring the latent environments.**
>
> **R1.1:** We would like to clarify the **motivations** for inferring the latent environment labels, which are in two folds. **(1)** Although invariant learning methods [1-4] have achieved satisfactory OOD generalization under distribution shifts, most existing methods cannot be directly applied to graphs. One of the main obstacles is that graph data usually comes from a mixture of latent environments without accurate environment labels [5, 6], while most existing invariant learning methods require multiple training environments with explicit environment labels. Inferring the latent environments is inevitable in bridging this gap. **(2)** Following the invariant learning literature [1-4] and recent work [5], we assume that the input graph consists of an invariant and variant subgraph, where the invariant subgraph captures invariant relationships between predictive graph structural information and labels. The variant subgraph in turn captures variant correlations under different distributions, which are environment-discriminative features, motivating us to adopt the variant subgraphs to infer the latent environments.
>
> The **significance** of inferring the latent environment is reflected in two aspects. **(1)** Since our model can automatically infer the environment label of graphs without supervision, we can study invariant learning for graph representation learning under a mixture of latent environments.  We also further propose a theoretically-guaranteed model and achieve substantial performance gains on several synthetic and real-world datasets. **(2)**  The environment inference module utilizes the variant subgraphs, which can also promote the accurate identification of invariant subgraphs for better OOD generalization (as analyzed in Section 5.4).
>
> + **Q1.2. There lacks instruction to extract, utilize, and analyze the latent environment labels.**
>
> **R1.2**: As discussed in Sections 3.2 and 3.3, we **extract** (infer) the latent environment labels by clustering the representations of all the variant subgraphs.  After obtaining the inferred environment labels, we **utilize** the inferred environment labels to optimize the objective Eq. (8) which encourages the output graph representations to be truly predictive to the labels and invariant across different environments.
>
> We also conduct some empirical **analyses** in the experiments. **(1)** We plot environment inference results on the synthetic dataset. Figure 5 shows that the variant subgraphs perfectly capture the environment-discriminate features and the latent environments behind graph data are accurately inferred. **(2)** We find that the environment inference module and invariant learning module can mutually enhance each other, reflected in Figure 4 which shows that the test accuracy and the clustering performance improve synchronously over training. **(3)** During rebuttal, **we further conduct experiments** to compare our original model with the ablated version namely removing the environment inference module. (Please kindly refer to the response to Q7 for the results.) We observe a significant performance drop of this ablated model, which well justifies the effectiveness of our designed environment inference module. Besides, **we also conduct more experiments on environment inference in Appendix E.7** of the revised paper to analyze this module in our model.

---

> > ### Author Response · Authors · 2022-08-02
> > **Response to Reviewer oeB7 (Part 2/4)**
> >
> > +  **Q2. The technical novelty and contributions of the paper are neutral.**
> >
> > **R2:** Thank you for this comment. We would like to clarify the  technical novelty and contributions of this work as follows. **(1)** We focus on a novel and challenging problem, i.e., learning invariant graph representations in a mixture of latent environments. To the best of our knowledge, this problem is not explored by the existing works and its formulation is also not well established in the literature due to the unique challenges for graphs under distribution shifts. **(2)** Since graph data usually comes from a mixture of latent environments and most existing invariance regularizers rely on accurate environment labels, they cannot be directly applied to graphs. It also remains a new research direction and is more challenging to identify the complex invariant patterns on graphs among latent environments. **(3)** We kindly disagree that our model is a simple combination of  the k-means clustering and invariance regularizer. The environment inference module and invariant learning module focus on two complementary parts of the input graphs, i.e., variant subgraphs and invariant subgraphs, respectively. Therefore, the two modules can mutually promote each other to identify more accurate invariant and variant patterns during training process. This is also a new insight for graph OOD generalization. **(4)**  We design a theoretically grounded learning scheme to find a maximal invariant subgraph generator for solving the graph OOD generalization problem. Our model achieves performance gains on both synthetic and real-world benchmarks with significant improvements against various baselines. We believe these technical contributions are non-trivial and have potential impacts to the community.
> >
> > + **Q3.1. The difference between the invariance property in Assumption 3.1 and the last condition of Theorem 4.1.**
> >
> > **R3.1:** Thanks for this insightful comment. First, we would like to clarify Assumption 3.1 and the last condition of Theorem 4.1. The invariance property in Assumption 3.1 is first introduced to solve OOD generalization problem by the invariant risk minimization (IRM) [1], which is also widely adopted by follow-up works [2-4]. It assumes that the input instance consists of invariant features whose relation to the label is stable among different environments and variant features whose relation to the label is sensitive to environment changes. Therefore, Assumption 3.1 focuses on **the relationship between data and labels**. This assumption can be reasonably introduced into graph-structure data, which starts a new research direction for handling graph OOD generalization problems [5]. On the other hand, the last condition of Theorem 4.1 assumes there exists another environment $e^\prime$ where the distribution of the invariant subgraphs is the same as that in environment $e$. Therefore, it focuses on **the data distribution and the diversity of environments (not involving labels)**. Considering the above differences, they are fundamentally different assumptions.
> >
> > + **Q3.2. Whether these assumptions are hard to achieve.**
> >
> > **R3.2:**  We believe **both assumptions can be easily satisfied for real-world graphs**. For Assumption 3.1, it was reasonably introduced in [5] and can be verified for real-world graphs. For example, as present in Figure 1 in Appendix, for molecule graphs labeled by specific properties [6, 7], the optimal invariant subgraphs represent the "hydrophilic R-OH group"/"non-polar repeated ring structures", whose relationship with the label solubility/anti-solubility is truly predictive and invariant across different environments. And the corresponding variant subgraphs denote the shared carbon structure  or scaffold [6, 7], which could change across different environments.  Similarly, for superpixel graphs [5], invariant and variant subgraphs represent the edges corresponding to the digit itself and other edges from the background, respectively. The label is only dependent on the digit and the background information could change while having no influence on the labels. And for the Graph-SST2 datasets (as shown in Figure 7 in the main paper),  invariant subgraphs are the positive/negative words in the sentences that are salient for sentiment labels. For the assumption in Theorem 4.1, since we do not have ground-truth of environmental labels for graphs, we cannot easily visualize the results, but the diversity of environments is also commonly believed for graphs [5, 6]. Theorem 4.1 shows that theoretical OOD optimality can be achieved if the assumptions are satisfied.

---

> > > ### Author Response · Authors · 2022-08-02
> > > **Response to Reviewer oeB7 (Part 3/4)**
> > >
> > > +  **Q4. How do the $\rm GNN^\mathbf{M}$, $\rm GNN^\mathbf{V}$, and $\rm GNN^\mathbf{I}$ be jointly optimized?**
> > >
> > > **R4:** Thank you for this question. The pseudocode of our method in Appendix shows the details of the joint optimization process. Specifically, **(1)** we first decompose each input graph into an invariant and variant subgraph (i.e., $G_I$, $G_V$) by the mask matrix $\rm \mathbf{M}$ that is generated from the $\rm GNN^\mathbf{M}$ (line 3 in Algorithm 1). **(2)** Then we generate representations of the invariant and variant subgraph (i.e., $\mathbf{h}_I$ and $\mathbf{h}_V$) by $\rm GNN^\mathbf{I}$ and $\rm GNN^\mathbf{V}$ (line 5 and line 6 in Algorithm 1) respectively. Note that $\rm GNN^\mathbf{I}$ and $\rm GNN^\mathbf{V}$ adopt shared parameters (as presented in Appendix E.2 GNN Configurations). **(3)** Finally, we infer environments with clustering representations of variant subgraphs and calculate the invariant learning objective function (by Eq. (8)) to update all model parameters using back propagation. Therefore, the $\rm GNN^\mathbf{M}$, $\rm GNN^\mathbf{V}$, and $\rm GNN^\mathbf{I}$ can be jointly optimized.
> > >
> > > +  **Q5. Optimization of the adjacent matrixes $A_I$ and $A_V$.**
> > >
> > > **R5:** Thank you for this question. As stated in lines 135-138, directly optimizing a discrete matrix is indeed intractable in practice. Therefore, we follow DIR [5] to adopt a learnable GNN (denoted as $\rm GNN^\mathbf{M}$) to generate a **soft mask matrix** (Eq. (2)). And the soft mask value on an edge directly controls the message-passing strength between connected nodes (a very low strength means the edge barely passes any message). Finally, we can decompose the original graph into the invariant subgraph (i.e., $A_I$) and variant subgraph (i.e., $A_V$) (Eq. (3)) in an end-to-end manner.
> > >
> > > +  **Q6. More descriptions on the Silhouette score.**
> > >
> > > **R6:** Thank you for this suggestion. Silhouette score [8], a commonly used evaluation metric for clustering, is defined as the mean Silhouette coefficient over all samples. The Silhouette coefficient is calculated using the mean intra-cluster distance (denoted as $d_i$) and the mean nearest-cluster distance (denoted as $d_n$) for each sample. The Silhouette coefficient for a sample is $(d_n - d_i) / max(d_i, d_n)$. Therefore, **Silhouette score falls within the range $[-1, 1]$**. A silhouette score close to 1 means that the clusters become dense and nicely separated. The score close to 0 means that clusters are overlapping. And the score of smaller than 0 means that data belonging to clusters may be wrong/incorrect.  We have added these details on the Silhouette score in Appendix E.5 of the revised paper.
> > >
> > > In Figure 4, the Silhouette score in our experiments reaches approximately 0.75, which is consistent with the clustering performance in Figure 5. These results indicate that the environment inference module and invariant learning module can mutually enhance each other, leading to an accurate clustering performance and a promising OOD generalization ability.
> > >
> > > + **Q7.  The clustering cases without environment inference module.**
> > >
> > > **R7**: Thank you for this suggestion. Following your suggestion, we compare our original model (GIL) with an ablated version namely removing the environment inference module (termed as GIL w/o EI). So, for GIL w/o EI, the optimization objective in the invariant learning module (Eq. (8)) will use the randomly partitioned environments.
> > >
> > >  The results on the synthetic dataset SP-Motif ($r_{test}=1/3$) are as follows.
> > >
> > > | $r_{train}$ |   $r=1/3$    |   $r=0.5$    |   $r=0.6$    |   $r=0.7$    |   $r=0.8$    |   $r=0.9$    |
> > > | ----------- | :----------: | :----------: | :----------: | :----------: | :----------: | :----------: |
> > > | GIL         | 55.44 ± 3.11 | 54.56 ± 3.02 | 53.60 ± 4.82 | 53.12 ± 2.18 | 51.24 ± 3.88 | 46.04 ± 3.51 |
> > > |  GIL w/o EI  | 53.15 ± 2.31 | 51.25 ± 3.86 | 48.98 ± 6.15 | 43.18 ± 5.93 | 42.65 ± 4.51 | 39.15 ± 4.14 |
> > >
> > >
> > > We also conduct comparisons on the real-world datasets.
> > >
> > > |       |  MNIST-75sp  |  Graph-SST2  |   MOLSIDER   |    MOLHIV    |
> > > | ----- | :----------: | :----------: | :----------: | :----------: |
> > > | GIL   | 21.94 ± 0.38 | 83.44 ± 0.37 | 63.50 ± 0.57 | 79.08 ± 0.54 |
> > > | GIL w/o EI | 19.75 ± 1.98 | 81.98 ± 1.10 | 57.78 ± 1.21 | 75.81 ± 1.23 |
> > >
> > > From the results above, we can observe a significant performance drop under all comparisons when removing the environment inference module. It well justifies the effectiveness of our designed module, demonstrating that inferring environments with variant subgraphs can benefit learning invariant graph representations under a mixture of latent environments. We have added the analyses above into Appendix E.7.1 of the revised paper.

---

> > > > ### Author Response · Authors · 2022-08-02
> > > > **Response to Reviewer oeB7 (Part 4/4)**
> > > >
> > > > +  **Q8. Try different choices of  READOUT functions and GNN architectures.**
> > > >
> > > > **R8**: Thank you for this suggestion. As present in Appendix E.2 GNN Configurations, the READOUT functions and GNN backbones are set the same as one representative baseline DIR [5] for a fair comparison. Considering the significant improvements against baselines under the same setting, we think the effectiveness of the proposed model is justified. Following your suggestion, we also conduct additional experiments on two real-world datasets with different GNNs and readout functions.
> > > >
> > > > | Backbone and READOUT |     MOLSIDER     |      MOLHIV      |
> > > > | -------------------- | :--------------: | :--------------: |
> > > > | GIN + add pooling    |  63.50 ±  0.57   |   79.08 ± 0.54   |
> > > > | GIN + max pooling    |   63.37 ± 0.72   | **79.10 ± 0.42** |
> > > > | GIN + mean pooling   |   61.91 ± 0.75   |   78.16 ± 0.47   |
> > > > | GCN + add pooling    |   62.31 ± 1.12   |   77.23 ± 0.61   |
> > > > | GCN + max pooling    | **63.68 ± 0.91** |   77.61 ± 0.59   |
> > > > | GCN + mean pooling   |   61.33 ± 0.45   |   76.98 ± 0.36   |
> > > >
> > > > We can observe that  the choices of  READOUT functions and GNN architectures have a slight influence on the performances. Overall, our model is not very sensitive to their choices and can be compatible with most common READOUT functions and backbones. We have added the analyses above into Appendix E.8.1 of the revised paper.
> > > >
> > > >
> > > >
> > > > **References:**
> > > >
> > > > [1] Arjovsky et al., Invariant risk minimization. Arxiv, 2019.
> > > >
> > > > [2] Koyama et al., When is invariance useful in an Out-of-Distribution Generalization problem? Arxiv, 2020.
> > > >
> > > > [3] Chang et al.,  Invariant Rationalization. ICML, 2020.
> > > >
> > > > [4] Krueger et al., Out-of-Distribution Generalization via Risk Extrapolation (REx). ICML, 2021.
> > > >
> > > > [5] Wu et al., Discovering Invariant Rationales for Graph Neural Networks. ICLR, 2022.
> > > >
> > > > [6] Hu et al., Open Graph Benchmark. NeurIPS, 2020.
> > > >
> > > > [7] Duvenaud et al., Convolutional networks on graphs for learning molecular fingerprints. NeurIPS, 2015.
> > > >
> > > > [8] Peter J Rousseeuw, Silhouettes: a graphical aid to the interpretation and validation of cluster analysis. Journal of computational and applied mathematics, 1987.

---

> ### Author Response · Authors · 2022-08-08
> **Kind reminder to the reviewer**
>
> Dear Reviewer,
>
> We have provided detailed responses to your comments. We are wondering whether your concerns have been properly addressed.
>
> If you have further questions after reading the responses and the revised paper, it would be great to let us know. We are happy to address them.
>
>
>
> Best regards,
>
> The Authors

---

### Author Response · Authors · 2022-08-02
**General Response by Authors**

We would like to thank all the reviewers for their thoughtful suggestions on our paper. We are glad that the reviewers have some positive impressions of our work, including focusing on the new/important problem (oeB7, uGwK), clear presentations (52ur, uGwK), good mathematical groundings (oeB7), well-scoped evaluations (52ur), and significant improvements (oeB7, PXQD).

We have provided detailed responses to all the comments/questions point-by-point and also added new empirical evaluations. The summary of our updates is as follows:

- We further clarify the technical details and the applicability of the assumptions.
- We add more empirical analyses to justify the effectiveness of inferring latent environments (in Appendix E.7), including the ablation studies, directly using the ground-truth environments, different choices for the clustering algorithm, the mutual promotion of the designed modules in the additional dataset, etc.
- We add more baselines for comparisons and analyses (in Section 5).

The above updates are highlighted in the revision. We appreciate all reviewers’ time again. We are looking forward to your reply!

---

### Meta-Review · Area_Chair_U5Ub · 2022-08-24

**Recommendation:** Accept
**Confidence:** Certain

**Metareview:**

This paper focuses on a new research problem of learning invariant graph representations under distribution shifts, which considers the latent environment labels. The proposal is a joint learning framework called graph invariant learning (GIL), combing three different GNNs of various functions. The philosophy behind sounds quite interesting to me, namely, learning a maximally invariant graph predictor, which composes an environment inference module and an invariant subgraph identification module. The proposed method GIL has good empirical results on several datasets and related theoretical analyses, which further justify its effectiveness.

The clarity and novelty are clearly above the bar of NeurIPS. While the reviewers had some concerns on the significance and complexity, the authors did a particularly good job in their rebuttal. Thus, all of us have agreed to accept this paper for publication! Please include the additional experimental results in the next version.

**Award:**

No

---

### Decision · Program_Chairs · 2022-09-14

Accept